# The evolution of sexual signaling is linked to odorant receptor tuning in perfume-collecting orchid bees

Philipp Brand [1,6]*, Ismael A. Hinojosa-Díaz[2], Ricardo Ayala[3], Michael Daigle[1], Carmen L. Yurrita Obiols[4], Thomas Eltz[5] & Santiago R. Ramírez [1]*

Sexual signaling is an important reproductive barrier known to evolve early during the formation of new species, but the genetic mechanisms that facilitate the divergence of sexual signals remain elusive. Here we isolate a gene linked to the rapid evolution of a signaling trait in a pair of nascent neotropical orchid bee lineages, *Euglossa dilemma* and *E. viridissima*. Male orchid bees acquire chemical compounds from their environment to concoct species-specific perfumes to later expose during courtship. We find that the two lineages acquire chemically distinct perfumes and are reproductively isolated despite low levels of genome-wide differentiation. Remarkably, variation in perfume chemistry coincides with rapid divergence in few odorant receptor (OR) genes. Using functional assays, we demonstrate that the derived variant of *Or41* in *E. dilemma* is specific towards its species-specific major perfume compound, whereas the ancestral variant in *E. viridissima* is broadly tuned to multiple odorants. Our results show that OR evolution likely played a role in the divergence of sexual communication in natural populations.

[1] Department of Evolution and Ecology, Center for Population Biology, University of California, 1 Shields Avenue, 95616 Davis, California, USA. [2] Departamento de Zoología, Instituto de Biología, Universidad Nacional Autónoma de México, Tercer Circuito s/n Ciudad Universitaria Delegación Coyoacán, Apartado Postal 70-153, Ciudad de México 04510, Mexico. [3] Estación de Biología Chamela, Instituto de Biología, Universidad Nacional Autónoma de México, Apartado Postal 21, San Patricio, Jalisco 48980, Mexico. [4] Centro de Estudios Conservacionistas, Universidad de San Carlos de Guatemala, Avenida La Reforma, 0-63, Guatemala 01000, Guatemala. [5] Department of Animal Ecology, Evolution and Biodiversity, Ruhr University Bochum, Universitätsstrasse 150, 44801 Bochum, Germany. [6] Present address: Laboratory of Neurophysiology and Behavior, The Rockefeller University, 1230 York Avenue, 10065 New York, New York, USA. *email: pbrand@ucdavis.edu; sanram@ucdavis.edu

Sexual signaling allows organisms to detect, identify, and choose appropriate mating partners. The remarkable diversity of sexual communication systems found across the animal tree of life reflects the important role that pre-zygotic reproductive barriers have played in lineage diversification[1,2]. Minute differences in sexual signals can lead to reproductive isolation, enforcing species boundaries through the interplay of highly specific and reciprocally tuned signals, and sensory organs[3–5]. However, notwithstanding the important role that sexual signaling plays in the origin and maintenance of species boundaries, the genetic mechanisms underlying their evolution remain poorly understood. Chemical signaling—despite being the most ancient and widespread form of intersexual communication—has received limited attention[3].

Many animal species use chemical signals to identify and discriminate potential mates[6]. Similar to other pre-zygotic reproductive barriers, chemical signals are characterized by elevated rates of differentiation, especially among co-occurring sympatric lineages[2,4,7]. In addition, several gene families involved in the sensory perception of chemical signals have been shown to mirror these evolutionary dynamics[4,8,9]. For example, olfactory receptor genes tend to differentiate rapidly in disparate groups of animals including lemurs[10], rodents[11], and moths[12]. However, it remains unclear how sexual chemical signaling evolves in a genome-wide context and contributes to the early diversification of species.

Speciation theory predicts that genetic loci associated with the evolution of reproductive isolation should exhibit pronounced divergence relative to background genome-wide variation[13,14]. Accordingly, population-level analyses of lineages during the early stages of speciation provide excellent opportunities to determine the genetic basis of sexual communication and its role in the formation of new species. However, identifying loci associated with the evolution of sexual signaling and/or reproductive isolation is not a straightforward process. Genome-wide analyses of divergence often produce large numbers of candidate genes with unknown function[15–17], which are seldom functionally linked to relevant traits. To understand how genetic variation relates to the evolution of sexual signaling, it is necessary to establish a link between genotype and phenotype. Here we take this approach in a pair of incipient orchid bee lineages.

Orchid bees are one of the most important insect pollinators in the neotropical region[18]. Male orchid bees acquire chemical compounds from various environmental sources including orchid flowers, fungi, and rotten vegetation, and store them in highly specialized pouches in their hind tibiae[19]. Male bees release the resulting "perfume" bouquet in elaborate courtship displays at perching sites where mating takes place[20]. Although the exact function of perfume communication has not been demonstrated, behavioral[21], sensory[21–23], and macroevolutionary evidence[7,24] suggest that perfumes are important sexual signaling traits involved in pre-zygotic reproductive isolation, presumably by enabling species recognition. Recent broad-scale phylogenetic analyses revealed that perfume signals diverge rapidly, especially among species of relatively recent origin that co-occur in sympatry[7], a common signature of reproductive barrier traits. This is corroborated by species-specific neurophysiological responses and behavioral attraction towards conspecific perfume compounds[21], suggesting a highly specialized signaling function—the hallmark of sexual communication systems[6]. In this unique chemical communication system, the sense of smell is crucial for both perfume acquisition by males and perfume detection by females. Therefore, any genetic changes that modify the olfactory sensory perception (e.g. mutations in a single chemosensory gene) might have the potential to simultaneously introduce changes to male perfume signals and female perfume preferences[21], a scenario that can favor the rapid evolution of assortative mating[25].

*Euglossa dilemma* and *Euglossa viridissima* constitute a pair of recently diverged orchid bee lineages that diverged ~150,000 years ago[26]. Previous work conducted on the overlapping (sympatric) populations of these two lineages revealed discrete differences in male perfume collection behavior and perfume chemistry[21,26], in addition to rapid divergence in chemosensory receptor genes involved in olfactory detection[27,28]. Although previous work[27] showed that divergence in chemosensory genes is correlated with perfume differentiation, it remains unclear to what extent the evolution of specific chemosensory genes facilitated the evolution of perfume communication and reproductive isolation among these lineages. This species pair provides an excellent opportunity to identify the genetic basis and evolutionary mechanisms underlying the origin of reproductive isolation.

Here we use a population-level analysis of perfume chemistry and genome-wide genetic divergence throughout the entire geographic range of *E. dilemma* and *E. viridissima*, coupled with functional assays, to identify genetic loci associated with the evolution of perfume signaling in orchid bees. We find that the two lineages collect chemically distinct perfumes and are reproductively isolated, despite exhibiting low levels of genome-wide differentiation. We then show that variation in perfume chemistry between *E. dilemma* and *E. viridissima* coincides with two species-specific selective sweeps in different regions of the genome that likely evolved via strong positive selection. These two regions harbor two different tandem arrays of odorant receptor (OR) genes jointly containing 43 OR genes. Although most of the ORs in these regions exhibited low levels of divergence, we identified one receptor (Or41) that is significantly enriched with amino acid changes in *E. dilemma* but conserved in *E. viridissima*. Using functional assays, we demonstrate that the derived variant of Or41 in *E. dilemma* is specific towards its major species-specific perfume compound, whereas the ancestral variant in *E. viridissima* is broadly tuned to multiple odorant compounds. Taken together, our results suggest that divergence in chemical tuning in a key OR gene contributed to the evolution of pre-mating reproductive barriers in natural populations of these orchid bee lineages. This raises the intriguing possibility that genes controlling sensory perception might simultaneously drive the evolution of male traits and female preferences used in mate recognition, highlighting the importance of rapid pre-zygotic reproductive isolation in the formation of new species.

## Results

**E. dilemma and E. viridissima are genetically distinct.** To assess whether *E. dilemma* and *E. viridissima* are isolated genetic lineages, we first genotyped 232 males sampled across their geographic range in Central America (Fig. 1a, Supplementary Table 1, and Supplementary Data 1). A principal components analysis of genetic variance (PCA) based on 16,369 single-nucleotide polymorphisms (SNPs) revealed that the two lineages are genetically distinct in both allopatry and sympatry (Fig. 1b and Supplementary Fig. 1). *E. dilemma* and *E. viridissima* were not completely separated by a single PC axis (Fig. 1b and Supplementary Fig. 2), which is congruent with a scenario of incomplete lineage separation. This observation was supported by a genetic clustering analysis that first separated geographically distinct populations within *E. dilemma* before separating species (ADMIXTURE; Fig. 1c). The analysis also revealed the existence of three genetic lineages (Fig. 1c and Supplementary Fig. 3) with low differentiation (pairwise $F_{ST}$: 0.04–0.18; Supplementary Tables 2 and 3) including *E. viridissima* (Ev), a southern *E. dilemma* population ($Ed_{south}$), and a northern *E. dilemma* population ($Ed_{north}$). These results indicate that *E. dilemma* and *E. viridissima* form genetically distinct lineages with a complex evolutionary history.

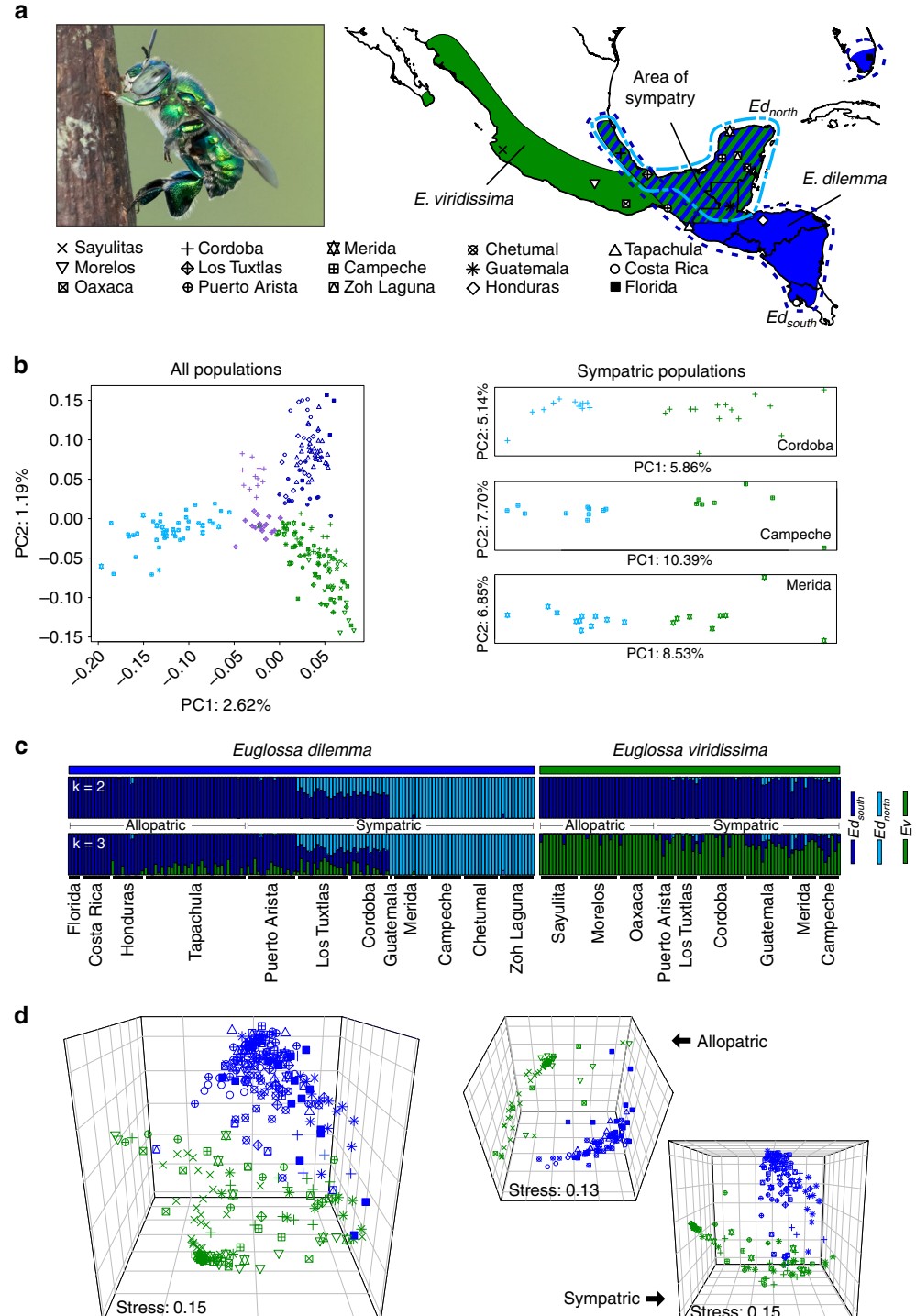

**Fig. 1 Genetic and phenotypic differentiation of *E. dilemma* and *E. viridissima*. a** Bees were collected at 15 sampling sites (Supplementary Table 1) including both allopatric (blue: *E. dilemma*, green: *E. viridissima*) and sympatric populations (hatched). Dashed lines indicate the distributions of *E. dilemma* genetic lineages following the coloring scheme described below. Photograph shows a *E. dilemma* male perching during perfume display. **b** *E. dilemma* (shades of blue) and *E. viridissima* (green) are genetically differentiated over the first two PC axes (left), which jointly explained <4% of the genetic variation, suggesting low genetic differentiation between lineages. Separation of the two lineages in sympatry suggests that they are reproductively isolated (right; Supplementary Fig. 1). Symbols are identical to legend in **a**. **c** Populations within *E. dilemma* (*k* = 2) were separated before species (*k* = 3) in a genetic clustering analysis, supporting population structure within *E. dilemma* (Supplementary Fig. 2) and low interspecific genetic differentiation. Several individuals drew ancestry from multiple genetic lineages suggesting admixture. **d** Perfume phenotypes were species-specific (left), independent of geography or co-occurrence (right), mainly due to the relative quantity of the major compounds HNDB and L97 (Supplementary Fig. 9). *E. dilemma* color scheme: light blue (**a**, **b**, **c**): *Ed_north*, dark blue (**a**, **b**, **c**): *Ed_south*, intermediate blue (**a**, **d**): *E. dilemma* (comprising all populations), purple (**b**): admixed populations Los Tuxtlas and Cordoba.

To assess lineage diversification among *E. dilemma* and *E. viridissima*, we further analyzed the above admixture patterns between $Ev$, $Ed_{south}$, and $Ed_{north}$. Several individuals drew ancestry from more than one genetic lineage, especially in sympatry (Fig. 1c), and an $f_4$-test demonstrated that the relationships between *E. dilemma* and *E. viridissima* cannot be explained by a simple bifurcating phylogeny ($f_4(Ed_{south}, Ed_{north}; Ev_{allopatric}, Ev_{sympatric})$: 0.001, z: 2.7, p-value = 0.007; Supplementary Table 4). This suggests that species differentiation might be affected by incomplete lineage sorting, gene flow upon secondary contact, or a combination of both. Indeed, demographic modeling supported a model of species differentiation (Akaike Information Criterion (AIC) weight = 1; Supplementary Table 5) where *E. viridissima* evolved from within *E. dilemma* with $Ev$ diverging from $Ed_{south}$ following the split of $Ed_{north}$ and $Ed_{south}$ (Supplementary Fig. 4). Further, the preferred model included secondary gene flow vertices among all three genetic lineages, consistent with a scenario of both incomplete lineage sorting and gene flow (Supplementary Fig. 4).

In addition to the genome-wide patterns of differentiation, a morphometric analysis of male mandible dentation, the only described diagnostic morphological trait for species identification[26], also supported a scenario of admixture among *E. dilemma* and *E. viridissima*. Consistent with previous observations[26], all *E. dilemma* males had three mandibular teeth, whereas most *E. viridissima* males had two teeth, but occasionally three. We found that in sympatric populations 26% of the *E. viridissima* males exhibited three mandibular teeth, whereas in allopatric populations only 3% of the males exhibited three teeth (Fisher's exact test, p = 0.0009; Supplementary Fig. 5). This pattern is consistent with the hypothesis that tooth number in *E. viridissima* may have resulted from recent introgression from *E. dilemma* in sympatry. Together, these results support a complex scenario of differentiation including incomplete lineage sorting and gene flow. In addition, our results support the hypothesis that *E. dilemma* and *E. viridissima* are distinct genetic lineages throughout their allopatric and sympatric geographic range.

**Perfume chemistry is species-specific.** Reproductive isolation is usually first established through the evolution of pre-mating barriers such as distinct chemical courtship signals[2,4]. Previous studies on sympatric populations suggest that variation in perfume signaling may contribute to reproductive isolation between *E. dilemma* and *E. viridissima*[21,26]. To determine whether perfume differentiation is species-specific, we conducted a distribution-wide analysis of perfume chemistry via gas chromatography–mass spectrometry (GC–MS) of 384 individuals (Supplementary Data 1). A non-metric multidimensional scaling (nMDS) analysis revealed strong differentiation of perfume composition into two distinct lineage-specific chemical phenotypes, independent of geography (Fig. 1d; analysis of similarity (ANOSIM) $R = 0.8$, $p = 0.001$; Supplementary Fig. 6). The most striking difference in perfume chemistry was the presence of two highly abundant lineage-specific compounds. HNDB (2-hydroxy-6-nona-1,3-dienyl-benzaldehyde[21]) was only present in perfume blends of *E. dilemma* and L97 (linoleic acid lactone-derivative[29]) was only present in perfume blends of *E. viridissima* (Supplementary Figs. 7 and 8, and Supplementary Table 6). These two diagnostic compounds accounted for the highest average proportion of overall perfume content per species (relative abundance HNDB: 55%, L97: 37%; Supplementary Table 7) and together contributed to 46.3% of the chemical differentiation (Similarity Percentage (SIMPER) analysis; Supplementary Table 8 and Supplementary Fig. 9). This result demonstrates that *E. dilemma* and *E. viridissima* have evolved lineage-specific signaling traits through discrete changes of the major perfume compounds.

**Selective sweeps linked to perfume differentiation.** We next performed a genome-wide scan of divergence between *E. dilemma* and *E. viridissima* to identify the genomic basis of perfume differentiation. We re-sequenced the genomes of 30 males from the three distinct genetic lineages ($n = 10$ for $Ed_{north}$, $Ed_{south}$, $Ev$; Supplementary Fig. 10 and Supplementary Data 1) and mapped them to the *E. dilemma* reference genome[30]. We capitalized on the fact that *E. dilemma* exhibits intraspecific population structure to identify genomic regions of high differentiation between *E. dilemma* and *E. viridissima* but not within *E. dilemma*. Therefore, we estimated net interspecific differentiation ($\Delta F_{ST}'$), which is calculated by subtracting the z-transformed intraspecific $F_{ST}$ ($F_{ST}'$) from the interspecific $F_{ST}'$, for non-overlapping 50 kb windows across the genome[31]. The resulting windows of elevated $\Delta F_{ST}'$ (>99th percentile) were clustered into seven distinct outlier peaks of varying size (0.05–1.7 Mb; Fig. 2b, Supplementary Fig. 11, and Supplementary Table 9).

We found that $\Delta F_{ST}'$ was correlated with gene density (Pearson's $r = 0.17$, $p = 0$), nucleotide diversity ($\pi$, $r < -0.24$, $p = 0$), absolute sequence divergence ($D_{xy}$, $r = -0.13$, $p = 0$), and linkage disequilibrium (LD; $r \geq 0.1$, $p = 0$; Supplementary Fig. 12), suggesting that regions with elevated $\Delta F_{ST}'$ are subject to indirect selection. These are commonly observed patterns of genome differentiation among species during the early stages of divergence and have been associated with a shared ancestral genomic landscape, purifying selection (i.e., background selection), divergent (positive) selection, and combinations thereof[32–37]. To isolate potential outlier windows evolving under divergent selection between *E. dilemma* and *E. viridissima*, we scanned for patterns of selective sweeps. Notwithstanding the general trend, we identified three outlier windows with elevated values of both $\Delta F_{ST}'$ and $D_{xy}$ (Fig. 2b), two of which also exhibited a highly skewed differential in nucleotide diversity towards *E. dilemma* ($\Delta \pi$; Fig. 2b), consistent with the hypothesis of a strong unilateral positive selection in this lineage. We identified signatures of an *E. dilemma*-specific selective sweep in one of these windows based on allele frequency spectra and haplotypes of the three distinct genetic lineages (Fig. 3a). Using this method on all outlier windows, we further identified one additional species-specific sweep in *E. viridissima* within another outlier region located on a different genomic scaffold (Fig. 3a and Supplementary Fig. 13), highlighting the presence of two different 50 kb windows containing locally restricted signatures of strong positive selection. This observation suggests that these genomic regions represent islands of divergence that likely harbor loci underlying reproductive isolation between *E. dilemma* and *E. viridissima*.

**Sweep regions are enriched for OR genes.** Close inspection of the two 50 kb windows with evidence of a selective sweep (Fig. 2a) revealed the presence of 14 genes in the *E. dilemma*-specific sweep window (Fig. 3a) and 4 genes in the *E. viridissima*-specific sweep window (Supplementary Fig. 14), all of which belong to the OR gene family. In both cases, the ORs are clustered into tandem arrays, of which the array in the *E. dilemma*-specific window exceeds the 50 kb window size and spans a total of ~170 kb and 39 ORs[28] (Fig. 3a). ORs belong to the largest chemosensory gene family in insects and are integral to the sensory detection of odorant compounds including pheromones[8]. As olfactory tuning is determined by the OR protein sequence, amino acid substitutions can lead to a shift in the odorant-binding properties of the receptor and thus modify sensory perception[5]. To identify the specific genetic targets of divergent selection, we mapped loci within these tandem arrays. We found that the regions containing the selective sweeps overlapped with both elevated interspecific $F_{ST}$ values (permutation test, $p = 0.0001$) and increased $D_{xy}$

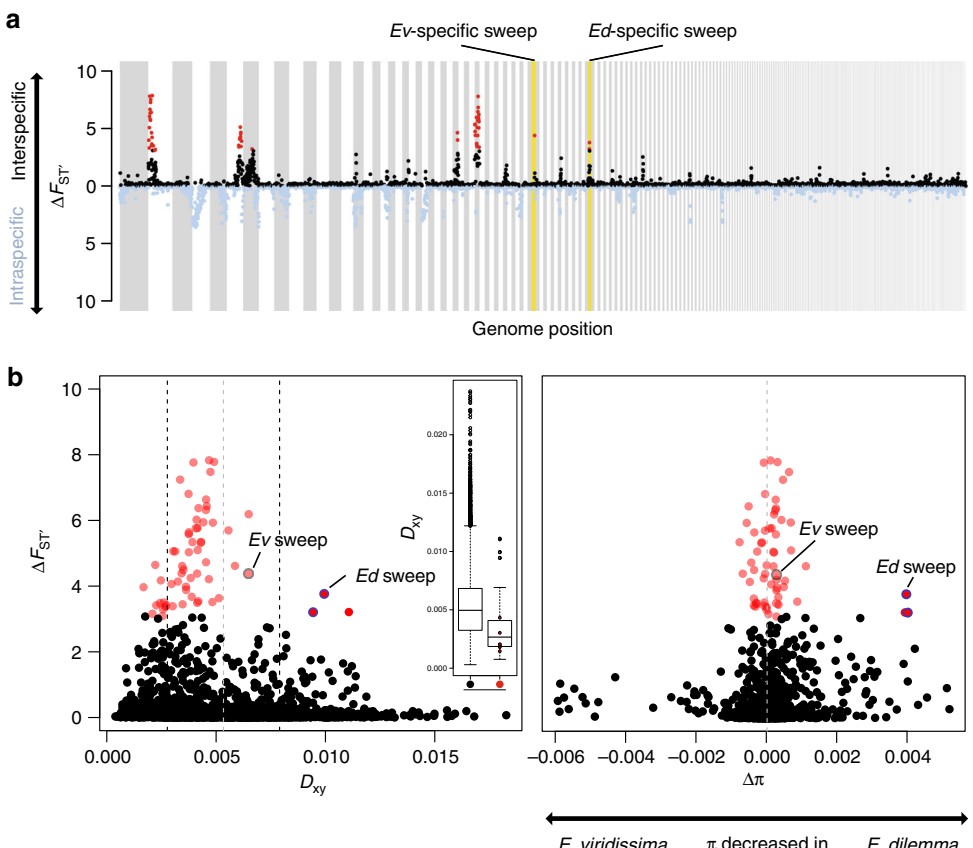

**Fig. 2 Whole-genome differentiation. a** Eight regions of the genome revealed higher interspecific (black) than intraspecific (blue) differentiation ($\Delta F_{ST}' >$ 99th percentile red). Outlier peaks with *E. dilemma* (*Ed*) and *E. viridissima* (*Ev*)-specific selective sweep signatures are highlighted in yellow. **b** Interspecific divergence ($D_{xy}$) was negatively correlated with $\Delta F_{ST}'$ (left panel, $r = -0.13$, $p = 0$) and significantly reduced in outlier regions (red) in comparison with non-outliers (black, inlet, Mann–Whitney *U*-test, $p = 0.001$, mean $D_{xy}$ of all seven outlier regions overlaid in red). Only two of the three $\Delta F_{ST}'$ outlier windows (circled blue) that revealed increased $D_{xy}$ also had a net differential of intraspecific nucleotide diversity ($\Delta \pi = \pi_{Ev} - \pi_{Ed}$) skewed towards *E. dilemma* (right panel), a pattern expected in genomic regions evolving under positive selection. Both correspond to the same outlier peak (rightmost of those with yellow background in **a**). Gray dashed lines: mean $D_{xy}$ and $\Delta \pi$. Black dashed lines: 1 SEM $D_{xy}$. Boxplot elements: center line represents the median, box bounds represent the 25th and 75th percentile, and whiskers represent the 25th percentile − 1.5 * the interquartile range and the 75th percentile + 1.5 * the interquartile range.

values (permutation test, $p = 0.0001$), and, in the *E. dilemma*-specific sweep, reduced nucleotide diversity ($\pi$) in *E. dilemma* (permutation test, $p = 0.0001$; Fig. 3a). Interestingly, the *E. dilemma*-specific sweep was centered around an OR gene, *OR41*, previously identified to evolve rapidly among these two lineages[27] (Fig. 3a). Indeed, of all the ORs in both tandem arrays, *Or41* was the only gene with an elevated number of non-synonymous substitutions (McDonald–Kreitman test, $p = 0.0048$; Supplementary Table 10), suggesting that this gene evolved under a strong positive selection, leading to changes in the amino acid sequence of the encoded receptor protein. A re-sequencing analysis of *Or41* ($n = 47$; Fig. 3b and Supplementary Tables 11 and 12) confirmed that the protein coding sequences were fixed for 19 substitutions between *E. dilemma* and *E. viridissima*, 17 of which were non-synonymous (Fig. 3c). A comparison with distantly related *Euglossa* species demonstrated that all fixed substitutions were derived (Supplementary Fig. 15) and evolved under a strong positive selection in *E. dilemma* ($d_N/d_S = 3.6$, $\chi^2 = 16.1$, $p < 0.0001$; Supplementary Table 13) but not *E. viridissima* ($d_N/d_S = 0.3$).

**Or41 variants differ in tuning towards perfume compounds.** To directly test the link between *Or41* and the observed specificity in perfume chemistry, we heterologously expressed the two *Or41*

variants present in *E. dilemma* and *E. viridissima* in the *Drosophila* olfactory T1 neuron lacking endogenous ORs[38] and measured electrophysiological responses to a diverse array of perfume compounds and odor blends (Fig. 4). We tested odors of potential ecological relevance in reproduction, nest construction, and foraging (all perfume compounds tested are commonly identified in floral odors with the exception of HNDB and L97). The conserved *Or41* variant present in *E. viridissima* did not show specificity towards a single compound but instead responded to various substances and mixtures, including waxes used in brood cell construction by females and several medium to long-chain fatty acids that are common in waxes[39]. Conversely, the derived *Or41* variant present in *E. dilemma* responded consistently to HNDB and *E. dilemma* perfume mixtures (which contain HNDB), but not to other odors (Fig. 4). This striking difference in olfactory tuning between *Or41* variants is likely caused by the positively selected non-synonymous changes that evolved in *E. dilemma*. Overall, this result suggests the transgression of an ancestral broadly tuned receptor variant to a derived highly specific receptor variant that responds only to HNDB.

## Discussion

Here we show that a simple phenotypic difference in a chemical signaling trait between lineages in the early stages of the

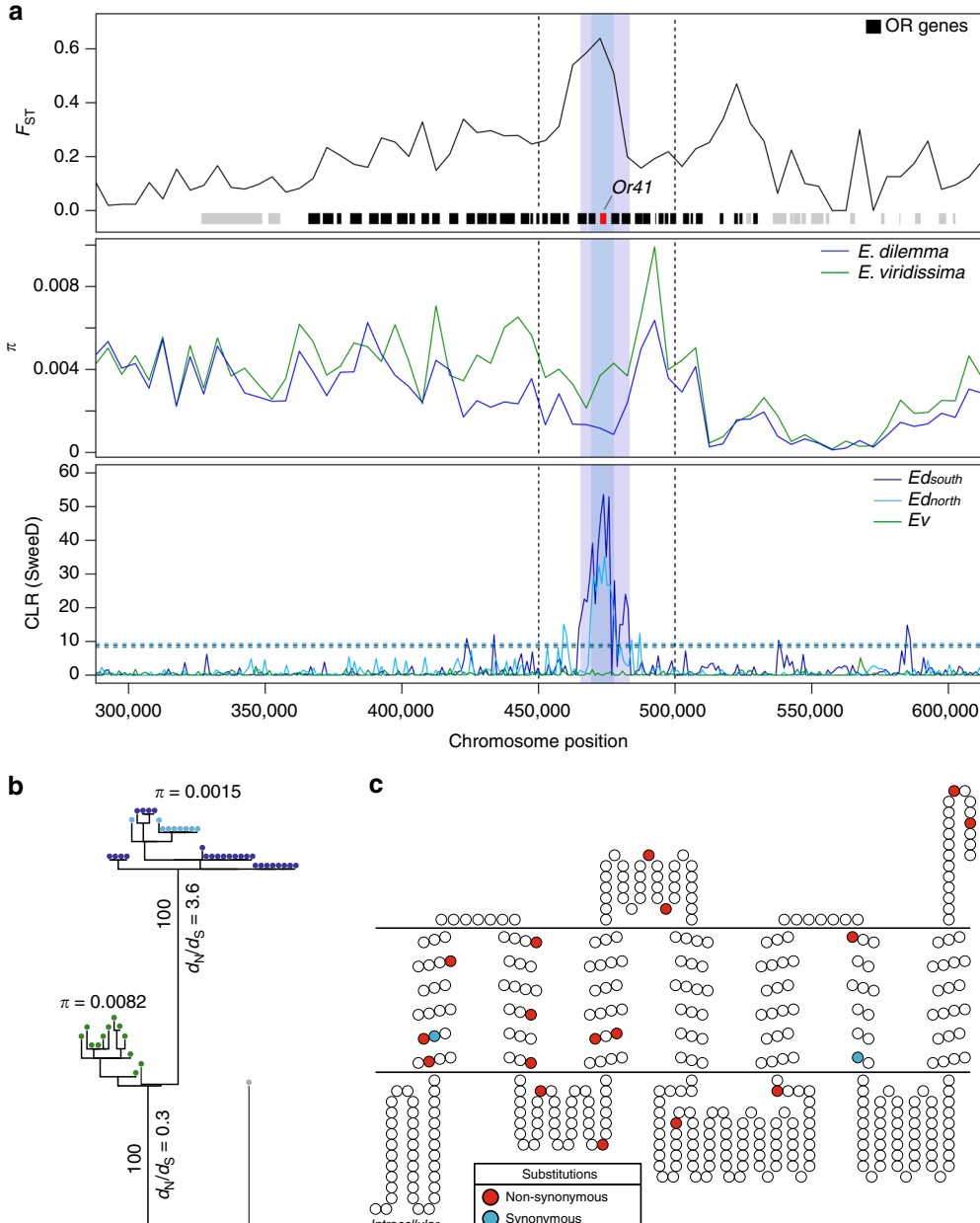

**Fig. 3 Odorant receptor (OR) gene *Or41* evolved through a species-specific selective sweep. a** The only *E. dilemma*-specific selective sweep identified was located within an $F_{ST}$ outlier window (dashed lines) overlapping with a high interspecific difference in $\pi$ within a tandem array containing 39 OR genes on scaffold_61 (green: *E. viridissima*; intermediate blue: *E. dilemma*). High composite likelihood ratios (CLR, bottom) in $Ed_{north}$ (light blue) and $Ed_{south}$ (dark blue) but not in $Ev$ (green) indicate a selective sweep shared by both *E. dilemma* lineages that overlap with *Or41* in the center (shaded regions). Horizontal dashed lines in the CLR panel indicate significance threshold for each lineage as indicated in the respective colors. **b** A maximum likelihood phylogeny of *Or41* ($n =$ 47 individuals) demonstrates that genotypes are species-specific. $\pi$ was five times lower in *E. dilemma* in comparison with *E. viridissima*. A $d_N/d_S$ analysis of species-specific genotypes with five outgroup species (gray dot) indicates positive selection on the *E. dilemma* branch ($d_N/d_S = 3.6$), but purifying selection of the ancestral genotype in *E. viridissima* ($d_N/d_S = 0.3$). Bootstrap support for tested branches is indicated. Color code as in **a**. **c** Seventeen of 19 substitutions mapped on the predicted membrane topology of the *Or41* protein were non-synonymous (red), whereas 2 were synonymous (blue).

speciation process is maintained despite complex lineage diversification, low genetic differentiation, and ongoing gene flow. Only strong selection can counteract the equalizing forces of admixture, highlighting the adaptive value of species-specific perfume compounds and their importance in reproductive isolation among orchid bees.

Our results suggest that *E. viridissima* likely evolved as a novel lineage from within one of two geographically distinct populations of *E. dilemma*. Our genetic, morphological, and chemical analyses demonstrate that *E. viridissima* represents an isolated

lineage across both allopatric and sympatric populations with *E. dilemma*, consistent with the hypothesis of *E. dilemma* and *E. viridissima* being reproductively isolated. In contrast, the two *E. dilemma* populations do not differ from each other in the morphological or chemical characters analyzed. This incongruity between reproductive isolation and phylogeny suggests that the two species diverged through paraphyletic speciation, which is considered common during the early stages of the speciation process but a transitory state in the majority of cases and thus rarely discovered[2,40]. Accordingly, although the evolutionary

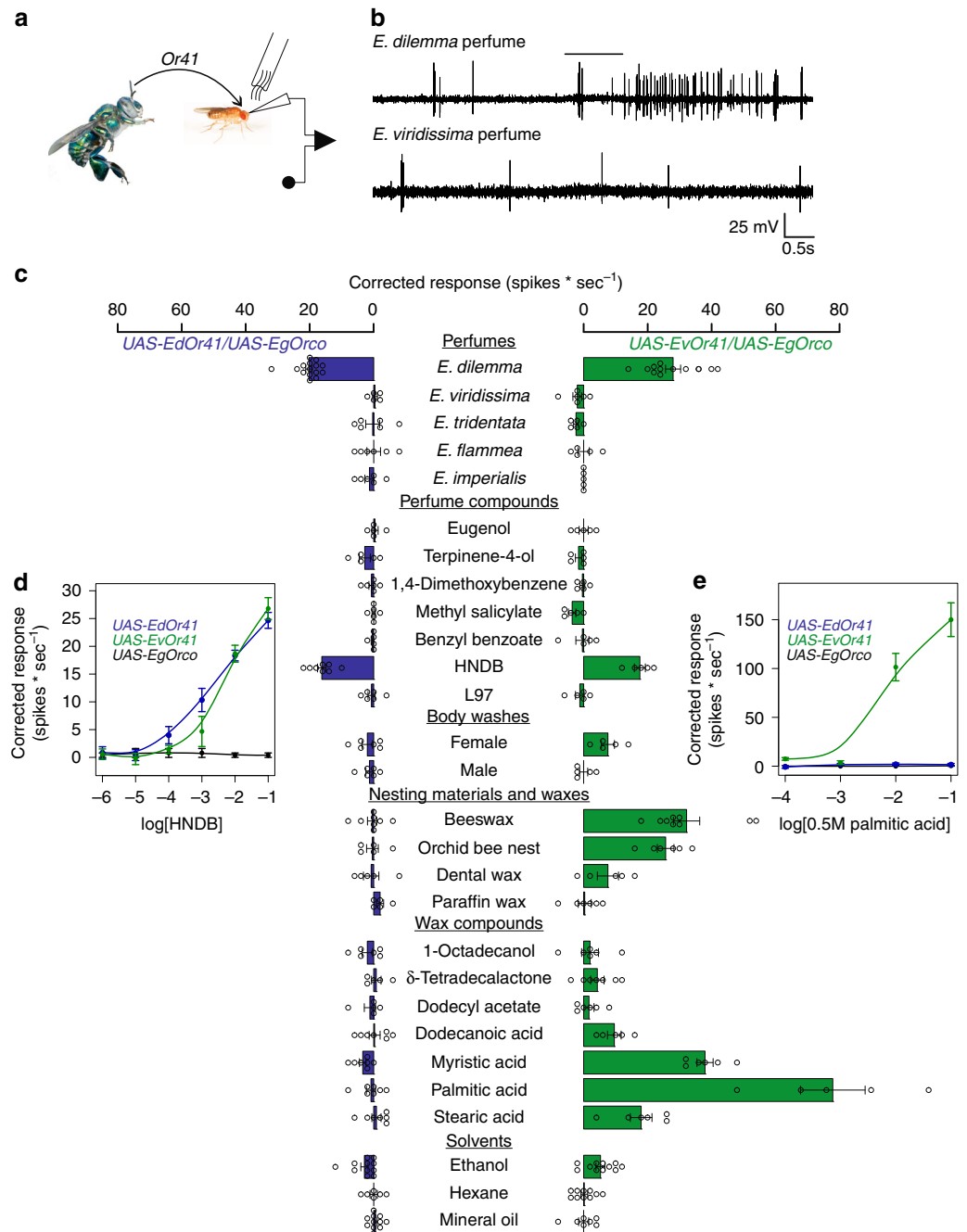

**Fig. 4 Or41 is narrowly tuned to HNDB in *E. dilemma*. a** Schematic of single sensillum recording. **b** Selective response of *E. dilemma* Or41 (*EdOr41*) to conspecific but not heterospecific perfume blend as indicated by an increase in spike frequency upon application of the odor stimulus (black bar). **c** The derived *EdOr41* variant (blue) responds specifically to HNDB, whereas the conserved *EvOr41* variant (green) responds to an array of odors and odor mixtures connected to natural waxes (mean ± SEM; *n* = 5–15). **d, e** Dose–response curves obtained from empty neurons expressing *EdOr41* (blue), *EvOr41* (green), or *EgOrco* (control, black) over *EgOrco* for HNDB (**d**) and palmitic acid (**e**). Pure compounds were diluted in mineral oil (1:100), 1,4-dimethoxybenzene, and fatty acids were diluted in ethanol (0.005 M), and all odor blends were extracted in hexane. Error bars represent the SEM. Source data are provided as a Source Data file.

history and geography of speciation in this group will require further investigation, it provides an ideal natural system to investigate the genetic basis of perfume signaling, an intriguing chemical communication system.

Although genome-wide analyses often lack resolution to identify loci that control barrier traits[36,37,41], we were able to identify and fully characterize the molecular and functional evolution associated with a genomic island of divergence. The only two species-specific selective sweeps we identified were located in regions enriched for OR genes. Further, we showed that in at least one of these regions an OR gene displayed signatures of species-specific divergent selection for amino acid changes. This provides strong evidence for an important role of OR gene divergence in the differentiation of *E. dilemma* and *E. viridissima*.

We identified a single OR gene that accumulated a surprisingly large number of non-synonymous substitutions. *Or41* differs in 17 non-synonymous substitutions between *E. dilemma* and *E. viridissima*, which is comparatively high for the ~150,000 years

divergence time estimated[26]. Thus, it is possible that this haplotype has not evolved de novo, but rather was present at low frequency in *E. dilemma* before the sweep. In addition, the haplotype under selection might have been acquired through introgression from more distantly related orchid bee species, which could have been swept through the populations. Similar scenarios have been identified in loci underlying wing patterning in *Heliconious* butterflies[42]. Regardless of the origin of this evolutionary change, our results show that the derived variant of *Or41* swept rapidly through *E. dilemma*, while the ancestral variant remained conserved in the *E. viridissima* lineage.

In this study we isolated several divergent loci that belong to the same gene family, thus opening a unique opportunity to understand the genomic landscape of pre-zygotic reproductive isolation at a fine genetic scale in natural populations of non-model organisms. Our findings provide strong evidence for the existence of genes underlying the evolution of a chemical signaling trait, including an olfactory receptor gene that evolved specificity for a signal-specific ligand in one of the daughter lineages. The genome-wide prominence of divergent OR evolution in combination with the functional link between *Or41* evolution and perfume differentiation raises the intriguing possibility that ORs might be essential for the speciation process in this system. Although additional data are needed to establish a causal link between the observed variation in *Or41* and reproductive isolation in natural populations, our results raise the possibility that genes controlling sensory perception might simultaneously drive the evolution of male traits and female preferences used in mate recognition, highlighting the importance of rapid pre-zygotic reproductive isolation in the formation of new species. This scenario is analogous to previous findings in *Laupala* crickets[43] and *Heliconious* butterflies[17] in which sexual signaling traits and mate preferences have been shown to evolve rapidly through linkage of the underlying genetic loci.

## Methods

**Sampling.** We sampled males of two orchid bee species *E. dilemma* and *E. viridissima* between 2014 and 2016 throughout the entire distribution area of each lineage (Supplementary Table 1) using chemical baits enclosed in tea strainers to prevent individuals to collect from them[27]. For species identification we used the number of mandibular teeth[26]. Sampling and export of bees was performed with the necessary permits issued to SRR (Costa Rica, permit 050-2013-STNAC by the Ministerio del Ambiente y Energía), IAH-D (Mexico, permit SGPA/DGVS/09586/15 by the Secretaría de Medio Ambiente y Recursos Naturales), and CLYO (Guatemala, permit 2756/2016 by the Consejo Nacional de Areas Protegidas). We dissected males in the field directly following collection and stored both hindlegs in 500 µL hexane for perfume extraction, whereas the rest of the body was preserved in 95% ethanol for subsequent morphological and genetic analyses. Perfume extracts and body tissue were stored at −20 °C until analyzed.

**Population genetics.** DNA was extracted from dissected flight musculature of tissue stored in 95% ethanol using the Blood and Tissue Kit (Qiagen). The extracted DNA was used for genotype-by-sequencing (GBS)[44]. Briefly, the DNA of each individual was digested using the EcoT22I restriction enzyme, followed by barcode ligation. Ninety-five individually barcoded samples were then pooled and PCR amplified, and finally size selected for ~300 bp using AMPure bead cleanup (Beckman Coulter, Brea, CA, USA). Libraries were then validated on a Bioanalyzer high-sensitivity DNA chip (Bio-Rad, Hercules, CA, USA) and sequenced on a HiSeq 2100 (Illumina, San Diego, CA, USA) in 150 bp single-end mode. Three individuals were run on each of the three lanes to correct for potential batch effects.

Sequencing reads were mapped to the *E. dilemma* reference genome[30]. As male Hymenoptera are haploid, we discarded all reads that mapped to more than one region of the genome to exclude repetitive loci. Nucleotides were called when a locus was sequenced in at least 50% of all individuals to a minimum read depth of four reads. SNPs were pruned if they were in LD of $r^2 \geq 0.2$ or had more than two alleles using plink[45]. After initial filtering, we excluded all individuals with <5000 called SNPs and repeated the filtering steps above on the final set of 232 individuals. The resulting pre-processed SNP set was then used for all downstream analyses.

To visualize genetic structure among populations and species, we performed a PCA in the R package SNPRelate[46]. Independent PCAs were performed on the entire set of individuals as well as on subsets of all sympatric sampling sites. In addition, we performed ancestry estimation in ADMIXTURE[47]. For both analyses, the SNP set was pruned to a minimum allele frequency (MAF) of 5%. To estimate admixture proportions for each individual in the dataset, we used ADMIXTURE based on SNPs called in at least 75% of individuals. We estimated levels of genetic structure for $k = 1$ to 10 jointly for all individuals. Each run was repeated ten times with random seeds including cross-validation (CV) error estimation for each level of $k$. We inferred the estimated number of genetic clusters using the mean CV error. We inferred the fixation index ($F_{ST}$) between genetic clusters in SNPRelate.

To test for "treeness" of the phylogeny including sympatric and allopatric populations of *E. dilemma* and *E. viridissima*, we used the $f_4$-test[48] as provided as part of the treemix software package[49]. The SNP set used was pruned to unlinked SNPs called in at least 75% of individuals with a MAF of 0.05 (1399 SNPs total).

We estimated the likelihood of 96 different demographic models of the 3 genetic lineages identified ($Ev$, $Ed_{north}$, and $Ed_{south}$) using Moments[50] based on the GBS dataset. The 96 demographic models tested included 16 different variations of bidirectional gene flow between all branches of six three-population models. As the phylogenetic relationships between the *E. dilemma* and *E. viridissima* lineages are unclear, we included models of all possible branching patterns ($16 \times 6$ models total). We kept the basic model as simple as possible with a single static population size per population (n1, n2, and n3) and two divergence times (t1 and t2) leading to a minimum of five estimated parameters (no migration/no gene flow). Models allowing for migration were incremental from single bidirectional vertices to up to four vertices (m1, m2, m3, and m4) added to the basic model leading to a maximum of nine estimated parameters. Parameter bounds were set to a minimum of 0.001 and a maximum of 100 for population size, 0 and 10 for divergence times, and 0 and 20 for migration rates. The likelihood of the empirical allele frequency spectrum (AFS) was estimated five times for each model. As likelihoods did not differ much between runs, we took the mean log likelihood for model comparison. To compare models, we used the AIC based on the log likelihoods of the empirical AFS given each of the 96 model AFS calculated in Moments. AIC scores, ΔAIC, and Aikaike weights were calculated in base R and used to identify the model with the best fit.

**Mandible morphometrics.** We determined the number of mandibular teeth in 414 males collected throughout the distribution range of *E. dilemma* and *E. viridissima* (Supplementary Data 1). To perform geometric morphometric analyses between *E. dilemma* and *E. viridissima*, we dissected the mandibles of 175 tridentate individuals (Supplementary Data 1) from the head capsule and mounted them on a paper point using clear nail polish. Mandibles were imaged using a Leica MZ 16A stereomicroscope with ×100 magnification. We captured stacked photographs with a JVC KY-F57U camera mounted on the stereomicroscope and merged them using Auto-Montage Pro (Synoptics, Cambridge, England). We converted the resulting images to the thine-plate spline format in tpsUtil, which we subsequently used to set five landmarks corresponding to the three tips of each tooth and the two indentations in-between teeth (Supplementary Fig. S8) in tpsDig. The resulting landmark data were analyzed with geomorph[51] in R by overlaying the landmarks of all individuals to identify species-specific geometric morphometric differences. A PCA of landmark shape variation was conducted using the plotTangentSpace function in geomorph.

**Perfume analysis.** We analyzed the perfume composition of 384 individuals (Supplementary Data 1) using GC–MS with an Agilent 7890B GC fitted with a 30 m × 0.25 mm × 0.25 µm HP-5 Ultra Inert column coupled with an Agilent 5977A MS (Agilent Technologies, Santa Clara, CA, USA). Using an autosampler, we injected 1 µL perfume extract splitless into the GC. Oven temperature was held at 60 °C for 3 min and then increased by 3 °C min$^{-1}$ until it reached 300 °C. Finally, the oven temperature was kept at 315 °C for 1 min. Both injector and transfer line temperatures were kept at 250 °C. Helium was used as carrier gas with a flow rate of 1.2 mL min$^{-1}$. Electron impact mass spectra were obtained by scanning between 30 and 550 mass-to-charge ratio (m/z). GC–MS data were processed using the MassHunter GC/MS Acquisition software vB.07.00 (Agilent Technologies) and analyzed in OpenChrom (Lablicate, Hamburg, Germany).

We created a manual mass spectral database from chromatogram peaks in OpenChrom, which we used to cross-reference the chromatograms of all analyzed individuals. To match peaks from different chromatograms, a minimum of 95% overlap of mass spectra with the manual database was required. We updated the database recursively as new compounds were detected. Individual compounds in the manual database were characterized by comparing mass spectra against the NIST05 database using the NIST MS Search software as well as other published mass spectra[26,29]. Chromatogram peaks were detected in OpenChrom using the first derivative peak detector with minimum signal-to-noise ratio set to 5 and a moving average window size set to 17. To determine total ion abundances of each peak, we integrated peaks in OpenChrom using the standard integrator. These steps were automatized to analyze all 385 samples in batch mode. Only peaks with an area ≥ 1% of the largest peak were included in downstream analyses. Peaks that corresponded to chemicals of endogenous origin such as cuticular hydrocarbons and other glandular secretions were identified via comparison with compounds in labial gland extracts, extracts of hindlegs of males hatched in captivity that had not collected perfume compounds, and via comparison with entries in previously

generated species-specific mass spectral databases[21]; these compounds were removed from all subsequent analyses.

We manually aligned the perfume profiles of all 384 individuals (Supplementary Data 1) using a combination of retention time and compound identity. The resulting matrix containing absolute quantities (total ion currents) for each compound was used for all downstream statistical analyses. Individuals with <10 collected compounds and thus in the early stages of perfume collection were discarded from subsequent analyses leading to a final set of 306 individuals in the perfume dataset. Similarly, compounds identified in less than three individuals were removed from the analysis, as these compounds likely represent molecules collected accidentally and are not likely involved in chemical signaling. We then transformed the absolute quantities of compounds to relative amounts per individual and analyzed the final perfume matrix in R. Therefore, we compared individual chemical profiles of the entire dataset as well as allopatric and sympatric subsets using three-dimensional nMDS analyses. Therefore, we calculated a triangular distance matrix between individuals using the Bray–Curtis (BC) index of dissimilarity, which is insensitive to compounds absent in sample pairs. Based on this BC matrix, we computed three-dimensional nMDS plots with 50 iterations per run using the ecodist package[52]. Each analysis was run ten times and convergence between runs was visually inspected. To statistically assess whether perfume profiles are more dissimilar between than within species, we conducted an ANOSIM test implemented in the vegan package[53]. We further estimated the relative contribution of each individual compound to the observed ordinal dissimilarities using the SIMPER method as implemented in vegan.

**Whole-genome analyses.** DNA from individuals with known GBS genotype was used for whole-genome library preparation using an adapted diluted Nextera DNA Sample Preparation procedure (New England Biolabs, Ipswich, MA, USA) to a minimum genome-wide read depth of 5×.

Reads were mapped to the *E. dilemma* reference genome[30] and SNPs were called with GATK[54]. Genome differentiation was analyzed in R using PopGenome[55]. We used a non-overlapping 50 kb sliding window approach to estimate pairwise relative genetic differentiation ($F_{ST}$), absolute sequence divergence ($D_{xy}$), and nucleotide diversity ($\pi$) for each window between all three genetic lineages. In addition, we calculated LD ($r^2$) within the same 50 kb windows using vcftools. Therefore, we considered only SNPs of at least 1 kb distance present in 90% of all individuals. Additional descriptive statistics were produced using plink[45] and base R.

We calculated the z-transformed net interspecific differentiation ($\Delta F_{ST}$)[31] by subtracting intraspecific diversity within *E. dilemma* from interspecific diversity between *E. dilemma* and *E. viridissima* ($\Delta F_{ST}$' = $F_{ST}$' [*Ev* vs. *Ed*] − $F_{ST}$' [*Ed_{north}* vs. *Ed_{south}*]). We then filtered the >99th percentile $\Delta F_{ST}$ regions as outliers of interspecific differentiation. To identify windows with $\pi$-values biased towards one species, we contrasted $\pi$ between species by subtracting $\pi_{Ed}$ from $\pi_{Ev}$ to calculate the net differential in intraspecific nucleotide diversity ($\Delta\pi$) between the two species. If $\Delta\pi$ equals 0, $\pi$ is indifferent between species in the corresponding genomic window.

We performed two independent tests for selective sweep signatures in all $\Delta F_{ST}$ outlier windows using (1) SweeD[56] and (2) hapFLK[57]. These two methods identify selective sweeps based on two different types of data including either the allele frequency spectra (SweeD) or haplotype information (hapFLK) of genomic regions. We ran hapFLK with $k = 5$ clusters and the default of 20 model fits (–nfit) on all scaffolds carrying $\Delta F_{ST}$ outlier windows. The optimal number of clusters was calculated using the fastPHASE CV method[58] as implemented in the imputeqc R package. SweeD was run on the folded site frequency spectra using a sliding window size of 1000 bp. The $\Delta F_{ST}$ outlier window that revealed a species-specific sweep pattern was then analyzed with a 10 kb non-overlapping sliding window in SNPrelate as described above. The significance value of the CLR statistic was calculated by simulating 50 kb regions with the evolutionary simulation framework SLiM[59] under the inferred demographic model. We simulated a total of one thousand 50 kb regions assuming a mutation rate of 3.5e−9 (mean of bumble bee[60] and honey bee[61]) and a recombination rate between 4.7e−8 (bumble bee[62]) and 2.6e−7 (honey bee[63]), which was randomly drawn from a uniform distribution throughout the simulated region. The simulation was performed for 600,000 generations with an assumed $N_E$ of 5,000 in the x-chromosome mode in SLiM, to simulate haplodiploid molecular population dynamics, and 10 male (i.e., haploid) individuals were drawn randomly at the end of each simulation. For each simulated dataset, we computed CLR test statistics in the same way as the empirical dataset. We then defined the significance threshold as the top CLR value for each lineage. Hapflk p-values were obtained by fitting a linear model to the distribution of estimated hapflk statistics using an M estimator as implemented in the R function "rlm"[57]. The resulting p-values were then adjusted for multiple testing using the Bonferroni method. We then performed permutation tests for $\Delta\pi$, $D_{xy}$, and $F_{ST}$ with 10,000 permutations in 50 kb windows across the two scaffolds with a selective sweep identified (scaffold_45 and scaffold_61) to test whether these significantly deviated from randomized population subsamples.

McDonald–Kreitman tests[64] of OR genes of the two tandem arrays with selective sweep patterns were performed by counting the number of polymorphic and fixed synonymous and non-synonymous of open reading frame (ORF) alignments of all individuals with whole-genome information and applying a four-field Fisher's exact test. Polymorphic sites were scored when present in more than one individual.

**Or41 evolutionary history.** We re-sequenced the *Or41* gene using tiled Sanger sequencing. We designed a set of four tiled PCR primer pairs spanning all 6 exons and 5 introns (Supplementary Table 12) to amplify the whole 2255 bp-long gene in a total of 47 individuals (12 *Ev*, 9 *Ed_{north}*, 26 *Ed_{south}*; Supplementary Table 13). Individual fragments were aligned to the *Or41* gene model derived from the *E. dilemma* reference genome[28,30] and the ORFs of either species[27] using mafft[65]. Based on these alignments, the *Or41* gene sequence of each individual was reconstructed in Geneious (Biomatters, San Francisco, CA, USA).

After the reconstruction of *Or41* genotypes, we produced a multi-sequence alignment including all individuals in mafft and analyzed it in MEGA[66]. We estimated $\pi$ within each species for the entire gene. Subsequently, substitutions in the ORF were identified visually and defined as fixed between species when all individuals of one species had a nucleotide different from all individuals of the other species. Otherwise, a substitution was defined polymorphic. Similarly, we visually identified substitutions as non-synonymous or synonymous based on the ORF. We then mapped the substitutions to the predicted membrane topology[27].

To reconstruct the *Or41* evolutionary history, we inferred a phylogenetic tree based on the entire ORFs of the 47 sequenced individuals and the publicly available ORFs of *Eacles imperialis*, *Eilema flammea*, and the more distantly related orchid bee *Eufriesea mexicana*[28] as outgroup. For tree inference, we estimated a maximum likelihood tree in RaxML[67], including 1000 bootstraps.

To test for selection along the branches leading to the respective species, we produced a gene phylogeny based on the consensus sequence for each of the two species together with *E. imperialis*, *E. flammea*, and *E. mexicana* as outgroup. The tree was used for a $d_N/d_S$ test using codeml in PAML[68]. Therefore, we estimated the likelihood of a model allowing for two or more $d_N/d_S$ values over branches in the tree with *E. dilemma* as foreground branch (M1), and a null model allowing only a single $d_N/d_S$ for all branches (M0). We conducted a likelihood ratio test, to test whether the model with branch variation is more likely than the null model ($\Delta = 2$ (ln(M1) − ln(M0)) with $\Delta$ approximating a $\chi^2$ distribution with one degree of freedom). Only fixed differences between *E. dilemma* and *E. viridissima* were taken into account to prevent overestimation of $d_N/d_S$ values.

**Functional analysis.** To amplify the target receptor alleles, we extracted antennal RNA from pools of 10–20 males of the same genotype (*Ev* or *Ed_{south}*) using the standard Trizol protocol (Thermo Fisher Scientific, Waltham, MA, USA) followed by first-strand cDNA synthesis with the SuperScript III kit (Invitrogen, Carlsbad, CA, USA). *EdOr41* and *EgOrco* (The Orco amino acid sequence is identical in *E. dilemma* and *E. viridissima*[27]) were amplified from *Ed_{south}* cDNA. *EvOr41* was amplified from *Ev* cDNA. PCR was conducted with the high-fidelity Phusion DNA polymerase system (New England Biolabs). The gel extracted PCR product was then ligated to the pUAST-attB vector[69]. Correct insertion and sequence identity was confirmed using Sanger sequencing and the final constructs were sent to BestGene (Chino Hills, CA, USA) for injection in *Drosophila melanogaster* eggs (attP40 stock) for targeted insertion on the second chromosome. Successful transformation was verified by a *white* marker gene on the injected construct, Sanger sequencing of attP target sites, and PCR of cDNA based on RNA extracted from 50 fly heads carrying the construct.

To drive the targeted expression of our UAS-receptor constructs in the olfactory sensory neuron of the T1 sensillum, we used the DmelOr67d-Gal4 empty neuron system[38]. Therefore, the UAS transgenes were crossed into a homozygous w-;UAS-receptor;DmelOR67d-Gal4 background. We performed one additional cross before single sensillum recording (SSR) leading to flies with the genotype w-; UAS-EgOrco/UAS-ExOr41; DmelOR67d-Gal4, co-expressing *EgOrco* and *Or41* of either *E. dilemma* or *E. viridissima* ("ExOr41"). Co-expression of *EgOrco* with *ExOr41* resulted in spontaneous activity and responsiveness to odors in an ExOr41-specific manner.

For SSR, 2- to 8-day-old female flies were trapped inside a 200 µl pipette tip so that the antennae and part of the eyes protruded from its narrow end. The tip was placed on a microscope slide using dental wax and the antenna was fixed on a glass slide covered with double-sided tape using a glass capillary. Using a micromanipulator (MPC-200, Sutter Instruments, Novato, CA, USA), an electrolysis-sharpened tungsten recording electrode was introduced at the base of a T1 sensillum using ×1000 magnification under an upright fixed-stage microscope (BX51Wi, Olympus, Tokyo, Japan) and the reference electrode was inserted into the eye or head of the fly. The recording electrode was connected to an amplifier (A-M Systems, Carlsborg, WA, USA) and the signals were fed through a Hum Bug noise eliminator (A-M Systems) into an analog–digital signal converter (BNC-2110, National Instruments, Austin, TX, USA). The signals were then visualized and recorded on a computer using WinEDR (Strathclyde). Odors and odor mixtures were delivered to the mounted fly via a humidified charcoal-filtered air stream (500 mL min⁻¹, CS-55, Ockenfels Syntech, Buchenbach, Germany). Test odors were loaded on a flint glass Pasteur pipet (10 µl for liquids, 10 mg for nesting materials and waxes) and heated for ~1 s with a handheld butane lighter to enable evaporation of chemicals[70], followed by delivery in 1 s puffs. Each odor was tested in 5–15 replicate sensillae with no more than three sensillae per fly. For dose–response curves, dilutions of a compound were tested on a single sensillum

per individual beginning with the solvent followed by the test compound in decreasing dilutions. Responses of T1 sensillae heterologously co-expressing *EdOr41*, *EvOr41*, or *EgOrco* with *EgOrco* were assessed by counting the number of spikes in the first 0.5 s following the onset of a puff and subtracting the number of spikes in the first 0.5 s preceding the onset of a puff and multiplied by 2 to calculate the corrected number of spikes per second in response to a given odor. T1 sensillae were identified by size, location, the presence of a single spike amplitude, and the absence of T4-specific response to methyl laurate. The DmelOR67d-Gal4/DmelOR67d-Gal4 genotype was confirmed by the absence of wild-type T1 response to cis-vaccenyl acetate.

The chemical compounds and compound mixtures that we tested were obtained from multiple sources. Perfumes mixtures were extracted from hindlegs of wild-caught males using 500 µl of hexane per leg, which we then pooled from 10 to 20 individuals. We tested a total of five species all of which occur in sympatry with *E. dilemma* and *E. viridissima*. Individually tested perfume compounds were purchased (≥95% purity, Sigma Aldrich, St. Louis, MO, USA) with the exception of HNDB and L97 (≥99% purity), which were isolated from crude tibial hexane extracts using preparative GC. Analytes were trapped inside short pieces of megabore column (DB-1) connected to the end of the chromatographic column (DB-5, 30 m, 0.53 mm ID, housed in a HP 5890 II GC). Trapping success, concentration, and purity of isolated compounds was confirmed by GC–MS using a HP 5890 II GC fitted with a 30 m nonpolar DB-5 column and a HP 5972 mass selective detector (Hewlett Packard, Wilmington, DE, USA). Body washes were produced by individually introducing the head, thorax (with legs removed in males), and abdomen of three freshly killed *E. dilemma* individuals of each sex into 500 µl hexane. After incubating for at least 10 min extracts were pooled to produce one male and one female whole-body wash. Nesting materials and waxes were purchased from various sources (beeswax: natural, Stakich, Inc., Troy, MI, USA; dental wax: white, Patterson Dental Company, St. Paul, MN, USA; paraffin wax: Gulf Wax®, Royal Oak Enterprises, Roswell, GA, USA) with the exception of nesting material, which was gathered freshly from an active *E. dilemma* nest. Pure compounds commonly found in natural waxes were purchased individually (≥95% purity, Sigma Aldrich). For SSR, all single compounds were diluted 1:100 in mineral oil (Sigma Aldrich) with the exception of 1,4-dimethoxybenzene and the fatty acids, which were diluted in pure ethanol (Sigma Aldrich) to a concentration of 0.005 M. Control chemicals were purchased in ≥95% purity (11-cis-vaccenyl acetate: Cayman Chemical, Ann Arbor, MI, USA; methyl laurate: Sigma Aldrich).

All experiments in this study were conducted in compliance with all relevant ethical regulations for animal testing and research.

**Reporting summary**. Further information on research design is available in the Nature Research Reporting Summary linked to this article.

## Data availability

All sequencing data were deposited in the sequence read archive of the National Center for Biotechnology Information (NCBI) database under BioProject number PRJNA529235. The *E. dilemma* genome assembly v1.0 is available from the i5k workspace at the National Agricultural Library [https://i5k.nal.usda.gov/euglossa-dilemma] and the NCBI under BioProject number PRJNA388474. Mandible morphology stacks and GC–MS data are available through Dryad [https://doi.org/10.5061/dryad.1g1jwstrf][71]. The source data underlying Figs. 4c, 4d, and 4e are provided as a Source Data file.

## Code availability

The custom scripts used to analyze and visualize the data are available on github [https://github.com/pbrec/popgen-popchem].

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

## Acknowledgements

We thank K.W. McCravy for sharing the Honduras bee samples; T. Pokorny, J.J.G. Quezada-Euan, R. Medina, M.A. Arteaga, and C. Pozo for help with fieldwork; J. Fong for help with morphometry; the Ward lab for support with imaging; and C.S. McBride, G. Pask, R. Butlin, C. Smadja, and the Ramírez lab for helpful discussions. The project was supported by a UC Mexus Dissertation Research Grant (P.B.), the David and Lucile Packard Foundation (S.R.R.), the NSF (S.R.R., DEB-1457753), and the DFG (T.E., EL 249/11).

## Author contributions

Conceptualization: P.B., T.E. and S.R.R. Fieldwork: P.B., S.R.R., I.H.-D., C.L.Y.O. and R.A. Data generation: P.B. - M.D. re-sequenced *Or41*. Project administration, data curation, formal analysis, investigation, visualization: P.B. Funding acquisition: P.B., T.E. and S.R.R. Writing—original draft: P.B. Writing—review and editing: P.B., T.E. and S.R.R. with input from all authors. Supervision: S.R.R.

## Competing interests

The authors declare no competing interests.
