## [Peer Review File · Nature Communications]

Reviewers' Comments:

Reviewer #1:

Remarks to the Author:

Brand et al investigate speciation in a fascinating study system with a very integrative approach. Two orchid bee species seem to have evolved through preferring different perfume cocktails, whereby the same gene that makes males of different species collect different perfumes also makes females of different species prefer different perfumes collected by the males. Therefore, divergence at a single gene could rapidly lead to prezygotic reproductive isolation. While the evidence for only a single gene driving speciation is disputable, their follow-up studies on their focal gene convincingly shows that this gene plays an important role in assortative mating. The F_{st} peak shows additional signature of selection and expressing the Or41 variants of the two species in *Drosophila* changes the response to perfume mixtures in the flies. I think this study has a lot of potential but still requires major improvements. With a stronger embedding into existing speciation literature and a few more tests and visualization improvements, this study will be highly impactful. I hope the authors find my comments useful to get it there.

The Introduction and Discussion are basically limited to orchid bees. To highlight the importance of this study, the authors need to put their study into context. Why would this finding be relevant to the broad speciation or evolutionary biologists community? How commonly does speciation involve just a single gene? What is the speciation theory behind this? Why is it even relevant how many genes are involved in speciation? How does this system compare to other cases of incipient speciation?

Having the sympatric and allopatric populations is a great setup that the authors should take more advantage of. Loci involved in reproductive isolation should show less gene flow between the species. Testing with the f_4 test and/or TWISST and/or f_d should show allow to test the absence of gene flow at Or41 and also at the other F_{st} peaks.

Could it be that the big peaks at the beginning of some scaffolds and the peaks covering entire scaffolds are actually part of the same chromosome and reflect big inversions? The finding of elevated linkage disequilibrium among the peaks (if I understand this correctly, L. 109) would support this idea. Clearer tests of LD among peaks would be interesting to see.

If it is true that *E. viridissima* and *E. dilemma* differ only in preference for perfume composition, how can the authors rule out that this is simply a polymorphism or divergent local adaptation e.g. due to different plant compositions in the Southern and Northern parts of Central America? An alternative explanation of the observed patterns would be that there are two morphs under divergent selection which are maintained in the area of sympatry by a balance between divergent selection and gene flow. Alternatively, the very divergent dilemma haplotype at the gene Or41 could have introgressed from another species in the South and now sweep through the geographic area from South to North. If the perfume cocktail preference is really the only trait that differs, these alternative explanations should be discussed. I do not think that these explanations are more likely than the interpretation of the authors but I think more discussion in this direction is needed.

The authors claim on L. 94 that the perfume composition is the only phenotypic trait that distinguishes the two lineages. However, there seems to be a clear difference between mandibular teeth number. This finding actually convinces me more that the *E. viridissima* and *E. dilemma* are indeed speciating and do not merely represent an intraspecific polymorphism. I do not think that the finding of an additional trait speaks against the prime role of Or41 in driving speciation. The authors should not dismiss this finding (Line: 94: "Remarkably, perfume composition is the only phenotypic trait that distinguishes these two lineages (Fig. S8, Supplement)").

The ADMIXTURE and PCA plots of the GBS data shows that the genetic differentiation within *E. dilemma* is actually greater than the genetic differentiation between the two species. This should also be discussed in more detail.

Ruling out F_{ST} outliers due to absence of increased d_{xy} is not correct in my opinion. Please read the very recent papers by Matthey-Doret & Whitlock, 2019, *MolEcol* and Stankowski et al., 2019, *Plos Biology*. Given the very recent divergent times between the bee lineages and the high levels of gene flow as shown in the low genome-wide differentiation, we would not expect d_{xy} to be increased in regions under divergent selection. The fact that the authors do find increased d_{xy} in one of the peaks may suggest that the haplotypes under divergent selection in the two species at that locus are in fact very old. The haplotype that swept through *E. dilemma* may be introgressed from a more divergent species. Or how else would such a divergent haplotype with so many differences from the ancestral haplotype evolve in such a short time period? Do the authors think that the two species were separated allopatrically for a long time before coming together and losing their genome-wide differentiation through gene flow? Or how else would the authors imagine that such divergent haplotype could evolve in sympatry?

It is almost impossible to find out the geographic distributions of the *E. dilemma* lineages. Highlighting these in Fig. 1 would be very helpful. Next, it would be good to use a consistent colouring scheme whereby the two lineages of *E. dilemma* are always the same two shades of dark and light blue and if they are both referred to together, they are intermediate blue.

Minor comments:

L. 10: A single gene causing speciation requires explanation. Write here that the male sexual trait and the female preference are both mediated through the same gene.

L. 60: Give exact p-value

Discuss the population in Florida. In Fig. 1c, Florida should be moved to the very left as it is allopatric. There is also a weird bar in the $K=2$ plot at Florida. The PCA plots in Fig. 1b need a legend.

Fig. 2a: mention the highlighted peak in the legend.

L. 111: please avoid the term "linked selection". Many readers will confuse it with background selection, whereas in fact any type of selection would lead to hitchhiking effects and thus count as "linked selection".

Reviewer #2:

Remarks to the Author:

In "An odorant receptor gene underlies reproductive isolation in perfume-collecting orchid bees," authors use a combination of population genomics, phenotypic analysis, and functional analysis to identify genetic loci underlying differences in perfume composition between a recently diverged pair of orchid bee sister species. First, using a reduced-representation sequencing approach and 232 males collected across the range of *E. dilemma* and *E. viridissima*, they demonstrate that these species are genetically differentiated in sympatry ($F_{ST} = 0.10$). This finding suggests that there is

some form of reproductive isolation. Second, the authors demonstrate that the two species differ in perfume chemistry. Third, using whole-genome resequencing data in a subset of samples, they identify eight regions of the genome for which there is elevated divergence between species relative to differentiation between populations of one of the species. Fourth, using additional selection analyses, the authors further narrow these 8 regions to a single region containing a cluster of 14 ORs belonging to a tandem array of 39. Within this cluster is a single OR with an unusually high nonsynonymous substitution rate. Fifth, moving forward with this candidate gene, the authors expressed the two alternative forms of OR41 in the *Drosophila* empty neuron system. This analysis detected differential tuning of these two ORs. Whereas the *E. dilemma*-specific (and derived) form of OR41 responded specifically to *E. dilemma* odors, the *E. viridissima* form was more broadly tuned. Based on these findings, they conclude they have identified a speciation gene that can simultaneously drive the evolution of male traits and female preferences, leading to the rapid evolution of prezygotic isolation.

Overall, this is a well written paper filled with fascinating data collected in a promising system for studying the genetic basis of prezygotic isolation. I am especially impressed with how far these authors have dug into this charismatic, but very challenging non-model system. That said, the main conclusions of the paper are not supported by the data. With the right re-framing (and appropriate analysis methods--see comments below), I think these data could potentially be worthy of publication in a journal of this caliber. But as it stands, I do not find the story to be convincing. Below I outline where I think the data and analyses fall short of the claims being made and make some suggestions about how the manuscript could be improved. I recognize that this manuscript represents a tremendous amount of work and hope that these comments are helpful to the authors.

My primary concern regarding this manuscript is with the claims being made about links between this particular locus and reproductive isolation. These claims include (just to select a few examples):

- Title: "An odorant receptor gene underlies reproductive isolation"
- L8-10: "Here we show that a single gene underlies the evolution of reproductive isolation..."
- L18-19: "Our results show that reproductive isolation can evolve through divergence in a single gene regulating sexual communication..."
- L198-199: "Our findings provide strong evidence for the existence of a speciation gene..."
- L200-202: "The data presented here also support the hypothesis that a single gene controlling sensory perception can simultaneously drive the evolution of male traits and female preferences involved in mate recognition..."

In my opinion, none of these claims are supported by either these data or by these data in conjunction with data in other papers that are cited in the manuscript. While it is certainly plausible that there is a single chemosensory gene (possibly OR41) that simultaneously impacts male and female perfume preference, there isn't enough known about this system to support these bold claims. First, to my knowledge, there is no experimental evidence linking perfume composition to RI between this species pair. There are a couple of papers cited in support of this claim on L80 (refs 18 and 20) but these are not experimental studies. So functional links between RI and perfuming behavior are still highly speculative. Second, there is also nothing known (again, to my knowledge) regarding female perfume preferences in this species pair. Thus, even if males are differentiated by perfume, it is possible that the females do not have strong preferences (or that these are asymmetric, etc). Third, even if there are differences in both male perfume and female preference, there is certainly no requirement that male and female preferences must be genetically correlated--they certainly could be, but they need not be. Fourth, while OR41 resides in a region of the genome with some interesting patterns of variation, there are many other ORs in this region that weren't tested that could be important and several other interesting regions of the genome (some with arguably stronger signals of selection). In short, we cannot conclude anything about the genetic architecture of male perfuming behavior from these data (and even less about the genetic architecture of female mate preference and the relation

between the two). Fifth, while the functional tests of the different OR41 variants are interesting and impressive and certainly seem to indicate that the dilemma variant is more specific to dilemma smells, their relation to real behavioral variation in male and female bees cannot be inferred from these results. It is possible that this difference in tuning has no impact whatsoever on male perfume preferences.

While I understand that it is very challenging to establish causal links between genes and behavioral traits and reproductive isolation in a non-model system such as this, that is the evidence that is needed to support the bold claims that are made in this paper. I reiterate that this is a fascinating and impressive dataset--I would just encourage the authors to reframe this surrounding a question that can be adequately addressed with the data (and to be more transparent about the limitations of these data). For example, these data suggest that there is positive selection on a candidate gene underlying differences in perfuming behavior, which is incredibly diverse in orchid bees and possibly linked to speciation. That is still a very cool result even without the flashy single-gene speciation story.

That said, beyond my concerns about the overall framing and conclusions of this paper, I do have some concerns about the population genetic analyses and their interpretation that need to be addressed as well. These are outlined in more detail below.

While I agree that evidence of differentiation in sympatry is a good indicator that there is at least some reproductive isolation between the species, I find several aspects of this system and these results confusing and some of the findings seem circular. To combat these issues, I have several comments/recommendations:

- It is essential that the manuscript includes a clear description of how species were identified prior to downstream pop-gen and phenotypic analyses. For example, in L94-95, the authors state that "perfume composition is the only phenotypic trait that distinguishes these two lineages". But in L91-93, the authors state that this result demonstrates that *E. dilemma* and *E. viridissima* have evolved distinct perfume phenotypes." If perfume is the only diagnostic trait, were species IDed based on perfume? If so, the conclusion in L91-93 is circular (of course species are going to be differentiated in perfume if this is what was used to delimit them). Similar logic applies to the analysis of morphological characters in the supplemental methods (if tooth # was used to ID species, how can one analyze tooth number differences between species). If species were IDed based on clustering in pop gen analyses, that is obviously very problematic for conclusions regarding reproductive isolation (hopefully this was not the case). Species ID methods need to be made very explicit.

- It is very strange that what should be the most differentiated comparison (different species in allopatry: *Ev* vs. *Ed*-south; $F_{st} = 0.04$, Table S6) is the least differentiated of any of the comparisons (*Ed*-north vs. *Ev* $F_{st} = 0.10$; *Ed*-north vs. *Ed*-south $F_{st} = 0.09$). A model of divergence with some gene flow would predict that F_{st} between sympatric interspecific pairs would be reduced relative to allopatric interspecific pairs. This is not what the data show and this odd result needs further investigation and explanation.

- o Could it be a sampling artifact (i.e., computing F_{st} from combinations of samples from several different locations)? To evaluate this, it would be useful to include a supplemental table that reports F_{st} for all population pairs (inter- and intraspecific) for which there are sufficient samples.

- o Could it be a sequencing artifact? For example, due to uneven data missingness or batch effects? The authors mention something about controlling for batch effects among sequencing lanes (e.g., running 3 individuals on each of 3 lanes; Supplemental Materials L21-22), but do not discuss how they use this information. Nor do they discuss other potential sources of batch effects (e.g., batch effects during library preps). If samples from different populations and species were randomized across sequencing libraries, this is not so much of an issue, but that information should be provided.

- o Could it be due to additional cryptic species?
- o Could it be attributable to an unusual demographic scenario? Here, a demographic model that describes divergence history and timing/extent of gene flow during divergence would be informative. Demographic modelling (using genome-wide data) is also essential for proper interpretation of selection scans (see below).

I also had many concerns about the methods through which the authors zeroed in on a single candidate OR. These concerns include:

- While I agree with the logic of using F_{st} within a species to distinguish between F_{st} peaks between species that are due to reproductive isolation rather than differentiation arising via unrelated processes, the execution and interpretation of this analysis was somewhat different from the paper cited to justify use of this method (Vijay et al. 2016). First, in contrast to Vijay et al. 2016, these authors did not transform F_{st} to account for genome-wide variance in F_{st} . Second, regions contributing to RI would be expected to exhibit outlier status in BOTH F_{st} and ΔF_{st} (see Vijay et al. 2016). Here, the authors identified some candidate RI peaks (Including the candidate containing the OR cluster) that have pretty modest F_{st} differentiation overall, but elevated ΔF_{st} . To me, these could be explained as peaks that are under stabilizing selection across the range within a species (low intraspecific F_{st}), but not necessarily experiencing strong divergent selection between the species. This would be consistent with a scenario in which there is strong selection for specificity in dilemma (maybe due to different perfume availability across the range or other correlated ecological traits), but no selection for or against specificity in viridissima and a weak or non-existent reproductive barrier.
- L109-110: because LD estimates consist of many non-independent pairwise values, a Mann-Whitney U Test (which assumes independence among data points) is not an appropriate statistical test for comparing outlier LD vs. non-outlier LD. The one exception to this would be if individual windows were summarized by a single LD value (based on the reported p-value, this seems highly unlikely because there are only 8 outlier windows). To compare pairwise LD estimates obtained from outlier and non-outlier windows, a permutation test could be used.
- L111-113: again, with a sample size of 8 outlier windows, I do not see how a p-value of $P < 0.0001$ could be possible. It seems like something is amiss with this analysis leading to an inflation of significance (e.g., treating data points that are not independent as independent samples). Also, I do not see a figure 2c.
- L120-121: "We did not identify any additional species-specific sweeps..." In looking at Figure S11, there do appear to be at least some sweep-like peaks in other scaffolds. This needs some further explanation.
- For SweeD, no significance thresholds are given so it cannot be stated with confidence that there is a signature of selection (or not) via this method. To determine a significance threshold, it is necessary to first estimate a demographic model for the data, then perform neutral simulations under the inferred demographic model to obtain null SFS.
- Likewise, it is not clear how significance was determined for hapFLK. Again, using current best practices, significance thresholds for the HapFLK statistic should be obtained via running the software on neutral data simulated under an appropriate demographic model for the populations under consideration.
- L141-145, Fig 3a: "We found that the region containing the selective sweep overlapped with both elevated interspecific F_{st} , reduced nucleotide diversity in *E. dilemma* centered around a single OR gene". Without any sort of significance thresholds in this figure, this overlap is somewhat meaningless. Even if we accept these as sweep-like peaks (they very well could be), I don't see how this indicates OR41 alone. Sweep-like peaks need not be centered exactly on the target of selection. There are many other genes here--what do these look like in terms of dN/dS ? Is there any other reason why OR41 stands out?

Given the very low levels of divergence between these species, I find it very surprising that OR41 has

so many fixed differences (19 substitutions!) between the two species (and, if this is the case, am surprised that this region did not come out as a stronger differentiation outlier). 150,000 years is a very short time to fix that many AA substitutions in *E. dilemma*, leading me to think there is something else going on here. Given this unusual result, alternative explanations should be considered and ruled out, where appropriate. The two main alternatives that come to mind are (1) an introgressed OR from a distantly related *Euglossa* species or (2) the alternative ORs were paralogs rather than orthologs. With annotated ORs published for *Euglossa*, it should be fairly straightforward to confirm orthology. But the possibility that this is an introgressed OR should be considered.

For the OR41 functional tests, it would be useful to have more information regarding the choice of chemicals that were tested and the likelihood that bees would encounter these odors during normal perfume gathering. This would be helpful in interpreting the possible ecological significance of the broad vs. narrow tuning.

Reviewer #3:

Remarks to the Author:

This work examines a recent speciation event in orchid bees and the genes contributing to the reproductive isolation. I especially applaud the combination of many experimental approaches to examine this phenomenon. I was familiar with the bioRxiv preprint, and the addition of the Or41 characterization provides intriguing functional support for their genomic analysis (where many studies in the field stop at the identification of chemosensory genes without functional work).

I believe the work is strong and worthy of publication, and I only have one minor suggestion. I'd like to see some detail about the *Drosophila* expression system included in the main text or associated figure. Although this information about the at1 expression system is in the supplemental methods, any detail that would help those familiar with *Drosophila* heterologous expression systems.

Reviewers' comments:

Reviewer #1 (Remarks to the Author):

Brand et al investigate speciation in a fascinating study system with a very integrative approach. Two orchid bee species seem to have evolved through preferring different perfume cocktails, whereby the same gene that makes males of different species collect different perfumes also makes females of different species prefer different perfumes collected by the males. Therefore, divergence at a single gene could rapidly lead to prezygotic reproductive isolation. While the evidence for only a single gene driving speciation is disputable, their follow-up studies on their focal gene convincingly shows that this gene plays an important role in assortative mating. The Fst peak shows additional signature of selection and expressing the Or41 variants of the two species in *Drosophila* changes the response to perfume mixtures in the flies. I think this study has a lot of potential but still requires major improvements. With a stronger embedding into existing speciation literature and a few more tests and visualization improvements, this study will be highly impactful. I hope the authors find my comments useful to get it there.

Response: We found the comments and suggestions very useful and helpful.

The Introduction and Discussion are basically limited to orchid bees. To highlight the importance of this study, the authors need to put their study into context. Why would this finding be relevant to the broad speciation or evolutionary biologists community? How commonly does speciation involve just a single gene? What is the speciation theory behind this? Why is it even relevant how many genes are involved in speciation? How does this system compare to other cases of incipient speciation?

Response: We reframed the introduction and added more context including a general framework to contextualize our study in relation to the current theory on speciation. We also modified the text in the introduction to emphasize less on the single-speciation-genes and more on importance of understanding how sexual communication systems evolve to better understand their contribution to the evolution of reproductive isolation.

Having the sympatric and allopatric populations is a great setup that the authors should take more advantage of. Loci involved in reproductive isolation should show less gene flow between the species. Testing with the f_4 test and/or TWISST and/or f_d should show allow to test the absence of gene flow at Or41 and also at the other Fst peaks.

Response: We agree with the reviewer that the existence of allopatric/sympatric populations offer a great opportunity to perform tests for gene flow. We note that the interpretation of such analysis is not straightforward, given the supported scenario of paraphyletic divergence in dilemma/viridissima, and these tests require a monophyletic null-model. For example: ((Species1sym,Species1allo),(Species2sym,Species2allo)). Our data suggest the following relationships among populations: (Edil_sym,(Edil_allo,(Evir_sym,Evir_allo))). For this reason we think it is better to not include such test. We did, however, run an f_4 test for each 50KB window, which shows that there is no gene flow between E. dilemma and E. viridissima at $\Delta F_{st}'$ peaks. However, due to the nature of the ΔF_{st} analysis, this is not surprising, since it filters out regions with high Fst between but not within species. While this is a nice confirmation, we decided to not

include the f4 test in the manuscript, because it feels circular. In addition, our resequencing analysis of OR41 using Sanger re-sequencing indicated no introgression for this locus in the sympatric populations.

Could it be that the big peaks at the beginning of some scaffolds and the peaks covering entire scaffolds are actually part of the same chromosome and reflect big inversions? The finding of elevated linkage disequilibrium among the peaks (if I understand this correctly, L. 109) would support this idea. Clearer tests of LD among peaks would be interesting to see.

Response: We agree with the reviewer that it would be very interesting to see whether these peaks of divergence correspond to chromosomal inversions. Unfortunately, we do not have a linkage map available for either species. While LD is generally correlated with *Fst*, we did not see increased linkage in tests among outlier peaks based on our comparatively small whole-genome re-sequencing dataset (this might change with a higher number of individuals, though, since LD estimates are sensitive to *N*). We thus decided not to include this analysis, since it does not seem to be informative with regard to the focus of the manuscript.

If it is true that *E. viridissima* and *E. dilemma* differ only in preference for perfume composition, how can the authors rule out that this is simply a polymorphism or divergent local adaptation e.g. due to different plant compositions in the Southern and Northern parts of Central America? An alternative explanation of the observed patterns would be that there are two morphs under divergent selection which are maintained in the area of sympatry by a balance between divergent selection and gene flow. Alternatively, the very divergent *dilemma* haplotype at the gene *Or41* could have introgressed from another species in the South and now sweep through the geographic area from South to North. If the perfume cocktail preference is really the only trait that differs, these alternative explanations should be discussed. I do not think that these explanations are more likely than the interpretation of the authors but I think more discussion in this direction is needed.

We think that reviewer 1 is raising an important and interesting point here. However, the fact that *E. dilemma* and *E. viridissima* collect distinct perfumes in sympatry (Yucatan peninsula and Veracruz) does not lend support to the divergent local adaptation hypothesis. In addition, it has been shown previously that *E. dilemma* does collect species-specific perfume blends in different localities with high variability in flora and fauna (Ramirez et al. 2010, Chemical Ecology), which suggests that male orchid bees have the ability to acquire species-specific perfumes independently of the type of habitat they live in.

In addition, we note that that the perfumes are not the only traits that are differentiated between *E. dilemma* and *E. viridissima*. We have now clarified this in the manuscript by describing the divergence in tooth morphology. To make this more clear to the reader, we included a more substantial discussion of these points in the manuscript.

The authors claim on L. 94 that the perfume composition is the only phenotypic trait that distinguishes the two lineages. However, there seems to be a clear difference between mandibular teeth number. This finding actually convinces me more that the *E. viridissima* and *E. dilemma* are indeed speciating and do not merely represent an intraspecific polymorphism. I do not think that the finding of an additional trait speaks against the prime role of *Or41* in driving speciation. The authors should not dismiss this finding (Line: 94: "Remarkably, perfume composition is the only phenotypic trait that distinguishes these two lineages (Fig. S8, Supplement)").

Response: We followed this excellent suggestion and 'resurrected' our morphological analysis of the mandibles from the supplemental material. We

now describe our findings of morphological differentiation in more detail in the revised manuscript, and critically discuss this in the context of the other findings.

The ADMIXTURE and PCA plots of the GBS data shows that the genetic differentiation within *E. dilemma* is actually greater than the genetic differentiation between the two species. This should also be discussed in more detail.

Response: We now included a demography modeling analysis in the manuscript, which is consistent with a paraphyletic speciation scenario with *E. viridissima* originating from within *E. dilemma*. These results are concurrent with the ADMIXTURE and PCA results. We added a more detailed discussion to the manuscript.

Ruling out FST outliers due to absence of increased d_{xy} is not correct in my opinion. Please read the very recent papers by Matthey-Doret & Whitlock, 2019, *MolEcol* and Stankowski et al., 2019, *Plos Biology*.

Response: We agree with reviewer 1 here and we have now corrected this part of the manuscript.

Given the very recent divergent times between the bee lineages and the high levels of gene flow as shown in the low genome-wide differentiation, we would not expect d_{xy} to be increased in regions under divergent selection. The fact that the authors do find increased d_{xy} in one of the peaks may suggest that the haplotypes under divergent selection in the two species at that locus are in fact very old. The haplotype that swept through *E. dilemma* may be introgressed from a more divergent species. Or how else would such a divergent haplotype with so many differences from the ancestral haplotype evolve in such a short time period? Do the authors think that the two species were separated allopatrically for a long time before coming together and losing their genome-wide differentiation through gene flow? Or how else would the authors imagine that such divergent haplotype could evolve in sympatry?

Response: Following the suggestion by reviewer 1, we have now added a new section in the discussion in which we discuss the patterns of high divergence of Or41 haplotypes. The question of how these divergent haplotypes originated in the first place between these closely related species is interesting and it is possible that introgression of OR41 from other lineages (either now extinct or still extant) explains why we observe such an elevated divergence in the *E. dilemma* OR41 ortholog. However, testing these alternative scenarios falls beyond the scope of our study since properly testing alternatives scenarios would require a substantial whole-genome sequencing effort across several species in the phylogeny of *Euglossa*. We layout this as a possible scenario. We note that future studies may focus on testing these alternative scenarios with the appropriate genomic sampling.

It is almost impossible to find out the geographic distributions of the *E. dilemma* lineages. Highlighting these in Fig. 1 would be very helpful. Next, it would be good to use a consistent colouring scheme whereby the two lineages of *E. dilemma* are always the same two shades of dark and light blue and if they are both referred to together, they are intermediate blue.

Response: Done

Minor comments:

L. 10: A single gene causing speciation requires explanation. Write here that the male sexual trait and the female preference are both mediated through the same gene.

Response: We reframed the manuscript to take away our previous emphasis from the single speciation gene narrative. We point out the fact that male trait and female preferences can be influenced by changes in a single locus in the introduction and discussion.

L. 60: Give exact p-value

Response: Done.

Discuss the population in Florida. In Fig. 1c, Florida should be moved to the very left as it is allopatric. There is also a weird bar in the K=2 plot at Florida. The PCA plots in Fig. 1b need a legend.

Response: We agree with the reviewer that the Florida population is interesting due to its recent introduction. However, we think this part is of lower importance for the major focus of the manuscript and thus moved the discussion of the Florida population to the supplement. We moved the population to the left in Fig. 1c and removed the bar. The legend for the PCA plots is shown in Fig. 1a. We added a sentence to the figure legend to make this clearer.

Fig. 2a: mention the highlighted peak in the legend.

Response: Done. In addition, we labeled the selective sweep-containing sweeps in the figure.

L. 111: please avoid the term "linked selection". Many readers will confuse it with background selection, whereas in fact any type of selection would lead to hitchhiking effects and thus count as "linked selection".

Response: We removed the term from the manuscript.

Reviewer #2 (Remarks to the Author):

In "An odorant receptor gene underlies reproductive isolation in perfume-collecting orchid bees," authors use a combination of population genomics, phenotypic analysis, and functional analysis to identify genetic loci underlying differences in perfume composition between a recently diverged pair of orchid bee sister species. First, using a reduced-representation sequencing approach and 232 males collected across the range of *E. dilemma* and *E. viridissima*, they demonstrate that these species are genetically differentiated in sympatry ($F_{st} = 0.10$). This finding suggests that there is some form of reproductive isolation. Second, the authors demonstrate that the two species differ in perfume chemistry. Third, using whole-genome resequencing data in a subset of samples, they identify eight regions of the genome for which there is elevated divergence between species relative to differentiation between populations of one of the species. Fourth, using additional selection analyses, the authors further narrow these 8 regions to a single region containing a cluster of 14 ORs belonging to a tandem array of 39. Within this cluster is a single OR with an unusually high nonsynonymous substitution rate. Fifth, moving forward with this candidate gene, the authors expressed the two alternative forms of OR41 in the *Drosophila* empty neuron system. This analysis detected differential tuning of these two ORs. Whereas the *E. dilemma*-specific (and derived) form of OR41 responded specifically to *E. dilemma* odors, the *E. viridissima* form was more broadly tuned. Based on these findings, they conclude they have identified a speciation gene that can simultaneously drive the evolution of male traits and female preferences, leading to the rapid evolution of prezygotic isolation.

Overall, this is a well written paper filled with fascinating data collected in a promising system

for studying the genetic basis of prezygotic isolation. I am especially impressed with how far these authors have dug into this charismatic, but very challenging non-model system. That said, the main conclusions of the paper are not supported by the data. With the right re-framing (and appropriate analysis methods--see comments below), I think these data could potentially be worthy of publication in a journal of this caliber. But as it stands, I do not find the story to be convincing. Below I outline where I think the data and analyses fall short of the claims being made and make some suggestions about how the manuscript could be improved. I recognize that this manuscript represents a tremendous amount of work and hope that these comments are helpful to the authors.

My primary concern regarding this manuscript is with the claims being made about links between this particular locus and reproductive isolation. These claims include (just to select a few examples):

- Title: "An odorant receptor gene underlies reproductive isolation"
- L8-10: "Here we show that a single gene underlies the evolution of reproductive isolation..."
- L18-19: "Our results show that reproductive isolation can evolve through divergence in a single gene regulating sexual communication..."
- L198-199: "Our findings provide strong evidence for the existence of a speciation gene..."
- L200-202: "The data presented here also support the hypothesis that a single gene controlling sensory perception can simultaneously drive the evolution of male traits and female preferences involved in mate recognition..."

In my opinion, none of these claims are supported by either these data or by these data in conjunction with data in other papers that are cited in the manuscript. While it is certainly plausible that there is a single chemosensory gene (possibly OR41) that simultaneously impacts male and female perfume preference, there isn't enough known about this system to support these bold claims. First, to my knowledge, there is no experimental evidence linking perfume composition to RI between this species pair. There are a couple of papers cited in support of this claim on L80 (refs 18 and 20) but these are not experimental studies. So functional links between RI and perfuming behavior are still highly speculative. Second, there is also nothing known (again, to my knowledge) regarding female perfume preferences in this species pair. Thus, even if males are differentiated by perfume, it is possible that the females do not have strong preferences (or that these are asymmetric, etc). Third, even if there are differences in both male perfume and female preference, there is certainly no requirement that male and female preferences must be genetically correlated--they certainly could be, but they need not be. Fourth, while OR41 resides in a region of the genome with some interesting patterns of variation, there are many other ORs in this region that weren't tested that could be important and several other interesting regions of the genome (some with arguably stronger signals of selection). In short, we cannot conclude anything about the genetic architecture of male perfuming behavior from these data (and even less about the genetic architecture of female mate preference and the relation between the two). Fifth, while the functional tests of the different OR41 variants are interesting and impressive and certainly seem to indicate that the dilemma variant is more specific to dilemma smells, their relation to real behavioral variation in male and female bees cannot be inferred from these results. It is possible that this difference in tuning has no impact whatsoever on male perfume preferences.

While I understand that it is very challenging to establish causal links between genes and behavioral traits and reproductive isolation in a non-model system such as this, that is the evidence that is needed to support the bold claims that are made in this paper. I reiterate that this is a fascinating and impressive dataset--I would just encourage the authors to reframe this surrounding a question that can be adequately addressed with the data (and to be more transparent about the limitations of these data). For example, these data suggest that there is positive selection on a candidate gene underlying differences in perfuming behavior, which is incredibly diverse in orchid bees and possibly linked to speciation. That is still a very cool result even without the flashy single-gene speciation story.

Response: We agree with the points raised here by reviewer 2. We carefully followed each of the points raised by the reviewer here and reframed the manuscript (including title, abstract, introduction and discussion) so that our

conclusions better reflect the data. We agree our dataset cannot be used to directly demonstrate that perfume (and Or41) evolution drives the evolution of reproductive isolation between *E. dilemma* and *E. viridissima*. We have modified the language to reflect this.

In addition, upon reanalysis of our selective sweep scans, we identified a second outlier region. This region showed a significant sweep pattern in *E. viridissima* and intriguingly also contained only OR genes. While this supports the assessment of the reviewer that there is not enough support for the hypothesis of a 'single speciation gene', this finding strongly suggests that OR evolution plays an important role in the early evolution of the two species. In addition, our sequencing analysis and functional data identified a link between OR evolution and perfume chemistry, suggesting that OR evolution is intricately associated with the evolution of this sexual signaling system. We reframed the manuscript and focused the discussion on the evolution of signaling traits and the potential role they played in the speciation process.

That said, beyond my concerns about the overall framing and conclusions of this paper, I do have some concerns about the population genetic analyses and their interpretation that need to be addressed as well. These are outlined in more detail below.

While I agree that evidence of differentiation in sympatry is a good indicator that there is at least some reproductive isolation between the species, I find several aspects of this system and these results confusing and some of the findings seem circular. To combat these issues, I have several comments/recommendations:

- It is essential that the manuscript includes a clear description of how species were identified prior to downstream pop-gen and phenotypic analyses. For example, in L94-95, the authors state that "perfume composition is the only phenotypic trait that distinguishes these two lineages". But in L91-93, the authors state that this result demonstrates that *E. dilemma* and *E. viridissima* have evolved distinct perfume phenotypes." If perfume is the only diagnostic trait, were species IDed based on perfume? If so, the conclusion in L91-93 is circular (of course species are going to be differentiated in perfume if this is what was used to delimit them). Similar logic applies to the analysis of morphological characters in the supplemental methods (if tooth # was used to ID species, how can one analyze tooth number differences between species). If species were IDed based on clustering in pop gen analyses, that is obviously very problematic for conclusions regarding reproductive isolation (hopefully this was not the case). Species ID methods need to be made very explicit.

Response: We agree with the reviewer that our method of species identification has to be clear. We now added a description of how we identified the species in the field to the materials section. During field collections, we used the mandible morphology, which turned out to be highly reliable in allopatry and mostly reliable in sympatry. In the results we now report the divergence in the mandibular dentation trait explicitly. After perfume chemistry and GBS analyses, we identified that in sympatry a fraction of the bees identified as *E. dilemma* based on the mandibles were genetically and chemically more similar to *E. viridissima*. This variation in the morphological trait has been described before (Eltz et al. 2011, *Zoological Journal of the Linnean Society*), and we interpret it as likely caused by introgression from *E. dilemma* to *E. viridissima* in sympatry. In summary, we are using a hierarchy of data to assess species status of individuals: 1. Morphology, 2. Perfumes (based on perfume phenotypes found in unequivocally identifiable males based on mandibles), 3. Genetics (based on unequivocally identifiable males based on mandibles and perfume).

- It is very strange that what should be the most differentiated comparison (different species in allopatry: Ev vs. Ed-south; $F_{st} = 0.04$, Table S6) is the least differentiated of any of the comparisons (Ed-north vs. Ev $F_{st} = 0.10$; Ed-north vs. Ed-south $F_{st} = 0.09$). A model of

divergence with some gene flow would predict that F_{st} between sympatric interspecific pairs would be reduced relative to allopatric interspecific pairs. This is not what the data show and this odd result needs further investigation and explanation.

Response: We agree with the reviewer that these results seem counterintuitive at first. We further investigated the discrepancy in F_{st} and reproductive isolation (see below).

o Could it be a sampling artifact (i.e., computing F_{st} from combinations of samples from several different locations)? To evaluate this, it would be useful to include a supplemental table that reports F_{st} for all population pairs (inter- and intraspecific) for which there are sufficient samples.

Response: We now included pairwise inter- and intra-specific F_{st} estimates from our different sampling sites in the supplement (Table S4), which show similar results to the pairwise F_{st} s across subsamples. It is thus unlikely that our results are due to a sampling artifact.

o Could it be a sequencing artifact? For example, due to uneven data missingness or batch effects? The authors mention something about controlling for batch effects among sequencing lanes (e.g., running 3 individuals on each of 3 lanes; Supplemental Materials L21-22), but do not discuss how they use this information. Nor do they discuss other potential sources of batch effects (e.g., batch effects during library preps). If samples from different populations and species were randomized across sequencing libraries, this is not so much of an issue, but that information should be provided.

Response: This is a valid concern. In our library prep and sequencing we included individuals from all three genetic lineages (Ev, Edsouth, Ednorth) in all lanes, suggesting that our observations do not result from batch effects. The individuals we repeatedly ran on all 3 lanes furthermore were highly similar in the number of SNPs recovered and their position in the PCA and ADMIXTURE analyses (data not shown). In addition, we sequenced a different set of individuals from all 3 lineages in a single pilot run (not included in the analyses) using slightly different library prep, sequencing, and SNP calling protocols and recovered identical population relationships. Furthermore, our whole-genome re-sequencing shows similar patterns to the GBS analysis (Fig S10). Accordingly, the observations are highly unlikely to be a result of methodological artifacts.

o Could it be due to additional cryptic species?

Response: Our sampling is comprehensive across the geography for both chemical and genetic data. We think it is unlikely that there is another cryptic species within the two that we have described in this system. Reviewer 2 is probably thinking about E dilemma north and E dilemma south corresponding to two cryptic species (given the observed level of genetic divergence). However, our chemical, genetic and morphological data does not support such a scenario. In particular, we do not find a case where the two E dilemma lineages occur in sympatry. In the absence of strong evidence supporting a third cryptic species, we prefer to take the more conservative approach suggesting the existence of just two species. We also think that the unusual divergence within E dilemma and the close relationship between E viridissima and Ed south likely reflect a pattern of paraphyletic speciation and complex past demographic history. We also note that the current geographic distributions of these lineages are likely different from the past distributions.

o Could it be attributable to an unusual demographic scenario? Here, a demographic model that describes divergence history and timing/extent of gene flow during divergence would be informative. Demographic modelling (using genome-wide data) is also essential for proper

interpretation of selection scans (see below).

Response: This seems to be most likely the case. We conducted demography modeling based on Ev, Edsouth, and Ednorth, which suggested that Ev evolved from within E. dilemma, diverging from Edsouth after the Ednorth-Edsouth split. This lends support of a paraphyletic speciation scenario, which is deemed to be quite common in nature but ephemeral and thus hard to detect. We included the demography modeling analysis in the manuscript. Furthermore, we discuss these findings with respect to our other results in the manuscript.

I also had many concerns about the methods through which the authors zeroed in on a single candidate OR. These concerns include:

- While I agree with the logic of using F_{st} within a species to distinguish between F_{st} peaks between species that are due to reproductive isolation rather than differentiation arising via unrelated processes, the execution and interpretation of this analysis was somewhat different from the paper cited to justify use of this method (Vijay et al. 2016). First, in contrast to Vijay et al. 2016, these authors did not transform F_{st} to account for genome-wide variance in F_{st} . Second, regions contributing to RI would be expected to exhibit outlier status in BOTH F_{st} and ΔF_{st} (see Vijay et al. 2016).

Response: We now z-transformed our data following Vijay et al. 2016. The transformation did not change our results but lead to the removal of one of the 8 peak regions (not the one harboring Or41).

Here, the authors identified some candidate RI peaks (Including the candidate containing the OR cluster) that have pretty modest F_{st} differentiation overall, but elevated ΔF_{st} . To me, these could be explained as peaks that are under stabilizing selection across the range within a species (low intraspecific F_{st}), but not necessarily experiencing strong divergent selection between the species. This would be consistent with a scenario in which there is strong selection for specificity in dilemma (maybe due to different perfume availability across the range or other correlated ecological traits), but no selection for or against specificity in viridissima and a weak or non-existent reproductive barrier.

Response: This is a good and important point. These F_{st} patterns seem to be a matter of resolution. Our in-depth analysis of the 50kb region harboring OR41 shows that indeed both F_{st} and ΔF_{st} are highly elevated among species (now with support from permutation testing that we added to the revised manuscript). The pattern was much more pronounced when we reanalyzed the 50kb region with a window size of 5kb (Fig 3a, Fig S14). Ultimately, this makes sense, since the pattern of selection that we identified e.g. in the Or41-harboring scaffold are most likely restricted to a short region around the Or41 gene (the gene is ~2300bp of the 50kb region) and therefore we would expect less pronounced F_{st} outliers using a 50kb window size.

- L109-110: because LD estimates consist of many non-independent pairwise values, a Mann-Whitney U Test (which assumes independence among data points) is not an appropriate statistical test for comparing outlier LD vs. non-outlier LD. The one exception to this would be if individual windows were summarized by a single LD value (based on the reported p-value, this seems highly unlikely because there are only 8 outlier windows). To compare pairwise LD estimates obtained from outlier and non-outlier windows, a permutation test could be used.

Response: We removed this part from the manuscript and instead compared the correlation of LD and F_{st} across all 50kb windows throughout the genome. We find that LD is positively correlated with ΔF_{st} , supporting higher linkage in higher ΔF_{st} regions.

- L111-113: again, with a sample size of 8 outlier windows, I do not see how a p-value of $P < 0.0001$ could be possible. It seems like something is amiss with this analysis leading to an

inflation of significance (e.g., treating data points that are not independent as independent samples). Also, I do not see a figure 2c.

Response: We changed our analysis and only compared the non-outlier 50kb windows to the (now 7 - see above) outlier regions (as means of all 50kb windows included in each outlier region), to prevent treating 50kb windows within outlier regions as independent. The resulting p-value is 0.001. In addition, for more clarity, we overlaid the mean dxy values for the outlier regions on the boxplot in Fig 2b. We removed the erroneous reference to figure 2c.

- L120-121: "We did not identify any additional species-specific sweeps..." In looking at Figure S11, there do appear to be at least some sweep-like peaks in other scaffolds. This needs some further explanation.

Response: We thank reviewer 2 for noting this. By conducting additional analyses suggested by the reviewer, we identified a second significant species-specific selective sweep, which also harbored a tandem array of ORs. This lends even stronger support to the hypothesis that receptor evolution plays an important role in the early divergence of this system.

- For SweeD, no significance thresholds are given so it cannot be stated with confidence that there is a signature of selection (or not) via this method. To determine a significance threshold, it is necessary to first estimate a demographic model for the data, then perform neutral simulations under the inferred demographic model to obtain null SFS.

Response: We now estimated significance thresholds using demographic modeling.

- Likewise, it is not clear how significance was determined for hapFLK. Again, using current best practices, significance thresholds for the HapFLK statistic should be obtained via running the software on neutral data simulated under an appropriate demographic model for the populations under consideration.

Response: We followed the approach by the HapFLK programmers and determined the significance threshold from the chi-squared hapflk density.

- L141-145, Fig 3a: "We found that the region containing the selective sweep overlapped with both elevated interspecific F_{st} , reduced nucleotide diversity in *E. dilemma* centered around a single OR gene". Without any sort of significance thresholds in this figure, this overlap is somewhat meaningless. Even if we accept these as sweep-like peaks (they very well could be), I don't see how this indicates OR41 alone. Sweep-like peaks need not be centered exactly on the target of selection. There are many other genes here--what do these look like in terms of dN/dS ? Is there any other reason why OR41 stands out?

Response: We now include permutation tests to identify whether F_{st} , dxy , and $\Delta\pi$ values in the two selective sweeps are elevated, and found this to be highly significant for both sweep regions (except $\Delta\pi$, which was only significantly skewed towards the species with the sweep pattern in the *E. dilemma*-specific sweep). To test whether OR41 stands out, we added a McDonald-Kreitman test on all ORs in both tandem arrays, which shows that only for OR41 the number of fixed non-synonymous substitutions is significantly enriched supporting strong selection on this gene. Accordingly, OR41 stands out among the ORs in both tandem arrays.

Given the very low levels of divergence between these species, I find it very surprising that OR41 has so many fixed differences (19 substitutions!) between the two species (and, if this is the case, am surprised that this region did not come out as a stronger differentiation outlier).

150,000 years is a very short time to fix that many AA substitutions in E. dilemma, leading me to think there is something else going on here. Given this unusual result, alternative explanations should be considered and ruled out, where appropriate. The two main alternatives that come to mind are (1) an introgressed OR from a distantly related Euglossa species or (2) the alternative ORs were paralogs rather than orthologs. With annotated ORs published for Euglossa, it should be fairly straightforward to confirm orthology. But the possibility that this is an introgressed OR should be considered.

These are excellent points. We did not find any support that the Or41 variants are paralogs during our work on an earlier analysis of the OR gene family evolution using genomic data (Brand and Ramirez 2017), which was supported by our sanger re-sequencing of Or41. The sequence chromatograms never showed double peaks for any individual, suggesting that only one Or41 version (with identical flanking regions) exists in each species. It is possible that the derived Or41 haplotype is introgressed from another species, but our data does not allow reliable support for this scenario. However, we now explicitly discuss the possibility of an introgression scenario in the manuscript.

For the OR41 functional tests, it would be useful to have more information regarding the choice of chemicals that were tested and the likelihood that bees would encounter these odors during normal perfume gathering. This would be helpful in interpreting the possible ecological significance of the broad vs. narrow tuning.

Response: We now included this in the manuscript.

Reviewer #3 (Remarks to the Author):

This work examines a recent speciation event in orchid bees and the genes contributing to the reproductive isolation. I especially applaud the combination of many experimental approaches to examine this phenomenon. I was familiar with the bioRxiv preprint, and the addition of the Or41 characterization provides intriguing functional support for their genomic analysis (where many studies in the field stop at the identification of chemosensory genes without functional work).

I believe the work is strong and worthy of publication, and I only have one minor suggestion. I'd like to see some detail about the Drosophila expression system included in the main text or associated figure. Although this information about the at1 expression system is in the supplemental methods, any detail that would help those familiar with Drosophila heterologous expression systems.

Response: We included more details about the expression system in the main text, both in the Results and especially in the Methods section.

Reviewers' Comments:

Reviewer #1:

Remarks to the Author:

The authors have mostly implemented my comments well. The Introduction is now much improved and the figures are easier to read. I also appreciate the change of focus towards speciation in general and away from the one-allele mechanism. This integrative study combines many different methods and approaches to shed light on this exciting study system of speciation. I only have minor comments left.

The authors should consider making custom code publicly available. Particularly R code which is insufficiently explained such as the computation of deltaAIC and Aikaie weights of the demographic models.

Figure 1a: label the dashed outlines in the map

Figure 2: I cannot locate the two regions highlighted in panel a in the two b panels. This would be useful for comparison of the different lines of evidence of selection.

The Discussion could still be strengthened with more comparison to other systems with similar findings.

Reviewer #2:

Remarks to the Author:

I was Reviewer #2 on the previous version of this manuscript. Overall, I think the authors did a phenomenal job of addressing my concerns. I only noted two small issues that may require further editing/clarification prior to publication.

First, in the figure 2 legend, it may be helpful to clarify that $\Delta\pi$ is calculated as $\pi(\text{ev}) - \pi(\text{ed})$. It may also be helpful to add "-specific sweep" to the species names under the arrow in the last panel of this figure. Based on the arrow at the bottom of 2b, I think it is natural for the reader to want equate higher $\Delta\pi$ values as more nucleotide diversity in *E. dilemma*, which is the opposite of what is expected under a selective sweep in *E. dilemma*. The $\Delta\pi$ calculation is clarified in L445-L446 and indeed, higher values would represent a *E. dilemma* sweep, but it took me an embarrassingly long time to get this straight. Some simple changes to the figure legend and labels would prevent this misunderstanding.

Second, in this version of the manuscript, the authors are generally very careful regarding the conclusions they can draw from these data. The one exception is the very last sentence of the manuscript, in which the authors state that: "The data presented here also support the hypothesis that genes controlling sensory perception can simultaneously drive the evolution of male traits and female preferences..." Perhaps I have missed something, but I do not see how these data support that particular hypothesis. Yes, this paper provides evidence that Or41 differentiation is associated with perfume differences (as well as other traits that are differentiated in sympatry between these two species), with signatures of positive selection, and with differential tuning to odorants. And yes, it is possible that this differential tuning can simultaneously impact male perfuming behavior and female preferences. But again, this paper has not casually linked Or41 differences to male or female behaviors in these bees. I think it's fine to raise this intriguing possibility, but I do not think that the authors can make the claim that these data support that hypothesis. One possible edit would be

something along the lines of..."Although additional data are needed to establish causal links between Or41 variation and male and female traits, our data raise the intriguing possibility that..." Or something along those lines.

With these very minor edits, I think this paper would be a valuable contribution to Nature Communications.

REVIEWERS' COMMENTS:

Reviewer #1 (Remarks to the Author):

The authors have mostly implemented my comments well. The Introduction is now much improved and the figures are easier to read. I also appreciate the change of focus towards speciation in general and away from the one-allele mechanism. This integrative study combines many different methods and approaches to shed light on this exciting study system of speciation. I only have minor comments left.

The authors should consider making custom code publicly available. Particularly R code which is insufficiently explained such as the computation of deltaAIC and Aikaike weights of the demographic models.

We now made our custom code publicly available on github (<https://github.com/pbrec/popgen-popchem>). The repository includes code used for the GBS, perfume chemistry, genome-wide popgen, AIC, and electrophysiology data analysis as well as input files to execute the code.

Figure 1a: label the dashed outlines in the map

Done.

Figure 2: I cannot locate the two regions highlighted in panel a in the two b panels. This would be useful for comparison of the different lines of evidence of selection.

We modified Figure 2 to highlight the two regions from panel a in panel b.

The Discussion could still be strengthened with more comparison to other systems with similar findings.

We now modified the discussion to better compare our findings to other systems with rapidly evolving pre-zygotic mating barriers.

Reviewer #2 (Remarks to the Author):

I was Reviewer #2 on the previous version of this manuscript. Overall, I think the authors did a phenomenal job of addressing my concerns. I only noted two small issues that may require further editing/clarification prior to publication.

First, in the figure 2 legend, it may be helpful to clarify that $\Delta\pi$ is calculated as $\pi(\text{ev}) - \pi(\text{ed})$. It may also be helpful to add “-specific sweep” to the species names under the arrow in the last panel of this figure. Based on the arrow at the bottom of 2b, I think it is natural for the reader to want equate higher $\Delta\pi$ values as more nucleotide diversity in *E. dilemma*, which is the opposite of what is expected under a selective sweep in *E. dilemma*. The $\Delta\pi$ calculation is clarified in L445-L446 and indeed, higher values would represent a *E. dilemma* sweep, but it took me an embarrassingly long time to get this straight. Some simple changes to the figure legend and labels would prevent this misunderstanding.

We have now clarified Figure 2 by adding the equation of how we calculated $\Delta\pi$ to the figure legend and adding “ π decreased in” to the species names under the arrow. We did not add the reviewer’s suggestion (“-specific sweep”), because a difference in π between the two species in a given genetic region does not necessarily mean that it was caused by a selective sweep.

Second, in this version of the manuscript, the authors are generally very careful regarding the conclusions they can draw from these data. The one exception is the very last sentence of the manuscript, in which the authors state that: “The data presented here also support the hypothesis that genes controlling sensory perception can simultaneously drive the evolution of male traits and female preferences...” Perhaps I have missed something, but I do not see how these data support that particular hypothesis. Yes, this paper provides evidence that Or41 differentiation is associated with perfume differences (as well as other traits that are differentiated in sympatry between these two species), with signatures of positive selection, and with differential tuning to odorants. And yes, it is possible that this differential tuning can simultaneously impact male perfuming behavior and female preferences. But again, this paper has not casually linked Or41 differences to male or female behaviors in these bees. I think it’s fine to raise this intriguing possibility, but I do not think that the authors can make the claim that these data support that hypothesis. One possible edit would be something along the lines of...”Although additional data are needed to establish causal links between Or41 variation and male and female traits, our data raise the intriguing possibility that...” Or something along those lines.

We agree with the reviewer and modified the last sentence of the discussion as recommended so that it is clear that this is a hypothesis that we are proposing and a conclusion derived from the data.

With these very minor edits, I think this paper would be a valuable contribution to Nature Communications.